# Auto-Transfer: Learning to route transferable representations

**Keerthiram Murugesan**[1*]     **Vijay Sadashivaiah**[2*]     **Ronny Luss**[1]
**Karthikeyan Shanmugam**[1]     **Pin-Yu Chen**[1]     **Amit Dhurandhar**[1]

[1]IBM Research, Yorktown Heights     [2]Rensselaer Polytechnic Institute, New york

`keerthiram.murugesan@ibm.com  sadasv2@rpi.edu`
`rluss@us.ibm.com  karthikeyan.shanmugam2@ibm.com`
`pin-yu.chen@ibm.com  adhuran@us.ibm.com`

## Abstract

Knowledge transfer between heterogeneous source and target networks and tasks has received a lot of attention in recent times as large amounts of quality labelled data can be difficult to obtain in many applications. Existing approaches typically constrain the target deep neural network (DNN) feature representations to be close to the source DNNs feature representations, which can be limiting. We, in this paper, propose a novel adversarial multi-armed bandit approach which automatically learns to route source representations to appropriate target representations following which they are combined in meaningful ways to produce accurate target models. We see upwards of 5% accuracy improvements compared with the state-of-the-art knowledge transfer methods on four benchmark (target) image datasets CUB200, Stanford Dogs, MIT67 and Stanford40 where the source dataset is ImageNet. We qualitatively analyze the goodness of our transfer scheme by showing individual examples of the important features our target network focuses on in different layers compared with the (closest) competitors. We also observe that our improvement over other methods is higher for smaller target datasets making it an effective tool for small data applications that may benefit from transfer learning.[1]

## 1 Introduction

Deep learning models have become increasingly good at learning from large amounts of labeled data. However, it is often difficult and expensive to collect sufficient a amount of labeled data for training a deep neural network (DNN). In such scenarios, transfer learning (Pan & Yang, 2009) has emerged as one of the promising learning paradigms that have demonstrated impressive gains in several domains such as vision, natural language, speech, etc., and tasks such as image classification (Sun et al., 2017; Mahajan et al., 2018), object detection (Girshick, 2015; Ren et al., 2015), segmentation (Long et al., 2015; He et al., 2017), question answering (Min et al., 2017; Chung et al., 2017), and machine translation (Zoph et al., 2016; Wang et al., 2018). Transfer learning utilizes the knowledge from information-rich source tasks to learn a specific (often information-poor) target task.

There are several ways to transfer knowledge from source task to target task (Pan & Yang, 2009), but the most widely used approach is *fine-tuning* (Sharif Razavian et al., 2014) where the target DNN being trained is initialized with the weights/representations of a source (often large) DNN (e.g. ResNet (He et al., 2016)) that has been pre-trained on a large dataset (e.g. ImageNet (Deng et al., 2009)). In spite of its popularity, fine-tuning may not be ideal when the source and target tasks/networks are heterogeneous i.e. differing feature spaces or distributions (Ryu et al., 2020; Tsai et al., 2020). Additionally, the pretrained source network can get overwritten/forgotten which prevents its usage for multiple target tasks simultaneously. Among the myriad of other transfer techniques, the most popular approach involves matching the features of the output (or gradient of the output) of the target model to that of the source model (Jang et al., 2019; Li et al., 2018; Zagoruyko & Komodakis, 2016). In addition to the output features, a few methods attempt to match the features of intermediate states between the source and target models. Here, in this paper, we focus on the latter by guiding the target model with the intermediate source knowledge representations.

---

[*]Equal contribution, ordered alphabetically.
[1]Code available at `https://github.com/IBM/auto-transfer`

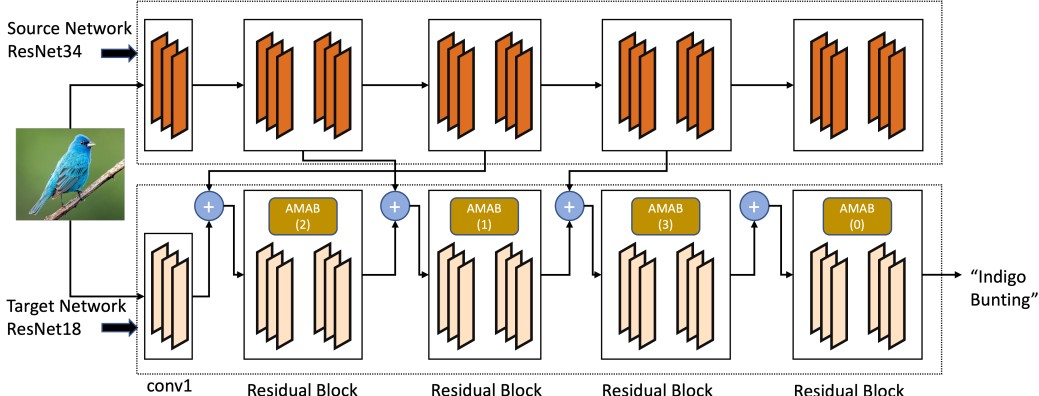

Figure 1: Illustration of our proposed approach. During training, an input image is first forward passed through the source network (such as ResNet34 trained on ImageNet) and the internal feature representations are saved. An adversarial multi-armed bandit (AMAB), for each layer of the target network (such as ResNet18), selects the useful source features (if any) to receive knowledge. Feature representations are then combined and fed into the next layer. In this example the following (target, source) pairs are selected: (1,2), (2,1), (3,3), (4, None). Parameters for AMAB and combination modules are optimized over training data. At test time, given an input image, representations mapping best feature representation between source-target layers, based on our method, are combined for the target network to make a decision.

While common approaches allow knowledge transfer between heterogeneous tasks/networks, it is also important to recognize that constraining the target DNN representations to be close to certain source DNN representations may be sub-optimal. For example, a source model, trained to classify cats vs dogs may be accessed at different levels to provide internal representations of tiger or wolf images to guide the target task in classifying tigers vs wolves. Since the source model is trained with a large number of parameters and labeled examples of cats and dogs, it will have learned several patterns that distinguish cat images from dog images. It is postulated that concepts or representations such as the shape of the tail, eyes, mouth, whiskers, fur, etc. are useful to differentiate them (Neyshabur et al., 2020), and it is further possible to reuse these learned patterns to generalize to new (related) tasks by accessing representations at the appropriate level. This example raises three important questions related to knowledge transfer between the source-target models: 1) *What* knowledge to transfer? 2) *Where* to transfer? 3) *How* to transfer the source knowledge?

While the *what* and *where* have been considered in prior literature (Rosenbaum et al., 2018; Jang et al., 2019), our work takes a novel and principled approach to the questions of *what*, *where* and *how* to transfer knowledge in the transfer learning paradigm. Specifically, and perhaps most importantly, we address the question of *how* to transfer knowledge, going beyond the standard matching techniques, and take the perspective that it might be best to let the target network decide what source knowledge is useful rather than overwriting one's knowledge to match the source representations. Figure 1 illustrates our approach to knowledge transfer where the question of *what* and *where* is addressed by an adversarial multi-armed bandit (*routing* function) and the *how* is addressed by an aggregation operation detailed later. In building towards these goals, we make the following contributions:

- We propose a transfer learning method that takes a novel and principled approach to automatically decide which source layers (if any) to receive knowledge from. To achieve this, we propose an adversarial multi-armed bandit (AMAB) to learn the parameters of our routing function.

- We propose to meaningfully combine feature representations received from the source network with the target network-generated feature representations. Among various aggregation operations that are considered, AMAB also plays a role in selecting the best one. This is in contrast with existing methods that force the target representation to be similar to source representation.

- Benefits of the proposed method are demonstrated on multiple datasets. Significant improvements are observed over seven existing benchmark transfer learning methods, particularly when the target dataset is small. For example, in our experiment on ImageNet-based transfer learning on the target Stanford 40 Actions dataset, our auto-transfer learning method achieved more than 15% improvement in accuracy over the best competitor.

## 2  RELATED WORK

Transfer learning from a pretrained source model is a well-known approach to handle target tasks with a limited label setup. A key aspect of our work is that we seek to transfer knowledge between heterogeneous DNNs and tasks. Recent work focused on feature and network weight *matching* to address this problem where the target network is constrained to be near the source network weights and/or feature maps. Network matching based on $L^2 - SP$ regularization penalizes the $\ell_2$ distance of the pretrained source network weights and weights of the target networks to restrict the search space of the target model and thereby hinder the generalization (Xuhong et al., 2018). Recent work (Li et al., 2018) has shown that it is better to regularize feature maps of the outer layers than the network weights and reweighting the important feature via attention. Furthermore, attention-based feature distillation and selection (AFDS) matches the features of the output of the convolutional layers between the source-target models and prunes the unimportant features for computational efficiency. Similar matching can also be applied to match the Jacobians (change in output with respect to input rather than matching the output) between source and target networks (Srinivas & Fleuret, 2018). Previous works (Dhurandhar et al., 2018; 2020) also suggested that rather than matching the output of a complex model, it could also be used to weight training examples of a smaller model.

Learning without forgetting (LwF) (Li & Hoiem, 2017) leverages the concept of distillation (Hinton et al., 2015) and takes it further by introducing the concept of stacking additional layers to the source network, retraining the new layers on the target task, and thus adapting to different source and target tasks. SpotTune (Guo et al., 2019) introduced an adaptive fine-tuning mechanism, where a policy network decides which parts of a network to freeze vs fine-tune. FitNet (Romero et al., 2014) introduced an alternative to fine-tuning, where the internal feature representations of teacher networks were used as a guide to training the student network by using $\ell_2$ matching loss between the two feature maps. Attention Transfer (AT) (Zagoruyko & Komodakis, 2016) used a similar approach to FitNet, except the matching loss was based on attention maps. The most relevant comparison to our work is that of Learning to Transfer (L2T-ww) (Jang et al., 2019), which matches source and target feature maps but uses a meta-learning based approach to learn weights for useful *pairs* of source-target layers for feature transfer. Unlike L2T-ww, our method uses a very different principled approach to combine the feature maps in a meaningful way (instead of feature matching) and let the target network decide what source knowledge is useful rather than overwriting one's knowledge to match the source representations. Finally, Ji et al. (2021) uses knowledge distillation based approach to transfer knowledge between source and target networks.

## 3  AUTO-TRANSFER METHOD

In this section, we describe our main algorithm for Auto-Transfer learning and explain in detail the adversarial bandit approach that dynamically chooses the best way to combine source and target representations in an online manner when the training of the target proceeds.

*What is the best way to train a target network such that it leverages pre-trained source representations speeding up training on the target task in terms of sample and time efficiency?* We propose a routing framework to answer this: At every target layer, we propose to route one of the source representations from different layers and combine it with a trainable operation (e.g. a weighted addition) such that the composite function can be trained together (see Figure 10 for an example of combined representations). We propose to use a bandit algorithm to make the routing/combination choices in an online manner, i.e. which source layer's representation to route to a given target layer and how to combine, while the training of the target network proceeds. The bandit algorithm intervenes once every epoch of training to make choices using rewards from evaluation of the combined network on a hold out set, while the latest choice made by the bandit is used by the training algorithm to update the target network parameters on the target task. We empirically show the benefit of this approach with other baselines on standard benchmarks. We now describe this framework of source-target representation transfer along with the online algorithm.

### 3.1  ROUTING REPRESENTATIONS

For a given image $x$, let $\{f_S^1(x), f_S^2(x), \cdots, f_S^N(x)\}$ and $\{f_T^1(x), f_T^2(x), \cdots, f_T^M(x)\}$ be the intermediate feature representations for image $x$ from the source and the target networks, respectively.

Let us assume the networks have trainable parameters $\mathcal{W}_S \in \mathbb{R}^{d_s}$ and $\mathcal{W}_T \in \mathbb{R}^{d_t}$ where $d_s$ and $d_t$ are the total number of trainable parameters of the networks. Clearly, the representations are a function of the trainable parameters of the respective networks. We assume that the source network is pre-trained. These representations could be the output of the convolutional or residual blocks of the source and target networks.

*Our Key Technique:* For the $i$-th target representation $f_T^i$, our proposed method a) maps $i$ to one of the $N$ intermediate source representations, $f_S^j$, or NULL (zero valued) representation; b) uses $T_j$, a trainable transformation of the representation $f_S^j$, to get $\tilde{f}_S^j$, i.e. $\tilde{f}_S^j(x) = T_j(f_S^j(x))$; and c) combines transformed source $\tilde{f}_S^j$ and the target representations $f_T^i$ using another trainable operation $\bigoplus$ chosen from a set of operations $\mathcal{M}$. Let $\mathcal{W}_{i,j}^{\bigoplus}$ be the set of trainable parameters associated with the operator chosen. We describe the various possible operations below. The target network uses the combined representation in place of the original $i$-th target representation:

$$\tilde{f}_T^i(x) = T_j(f_S^j(x)) \bigoplus f_T^i(x) \tag{1}$$

In the above equation, the trainable parameters of the operator depend on the $i$ and $j$ (that dependence is hidden for convenience in notation). The set of choices are discrete, that is, $\mathcal{P} = \{[N] \cup \text{NULL}\} \times \mathcal{M}$ where $[N]$ denotes set of $N$ source representations. Each choice has a set of trainable parameters $T_j, \mathcal{W}_{i,j}^{\bigoplus}$ in addition to the trainable parameters $\mathcal{W}_T$ of the target network.

## 3.2 LEARNING THE CHOICE THROUGH ADVERSARIAL BANDITS

To pick the source-target mapping and the operator choice, we propose an adversarial bandit-based online routing function (Auer et al., 2002) that picks one of the choices (with its own trainable parameters) containing information on *what*, *where* and *how* to transfer to the target representation $i$. Briefly, adversarial bandits choose actions $a_t$ from a discrete choice of actions at time $t$, and the environment presents an adversarial reward $r_t(a_t)$ for that choice. The bandit algorithm minimizes the regret with respect to the best action $a^*$ in hindsight. In our non-stationary problem setting, the knowledge transfer from the source model changes the best action (and the reward function) at every round as the target network adapts to this additional knowledge. This is the key reason to use adversarial bandits for making choices as it is agnostic to an action dependent adversary.

*Bandit Update:* We provide our main update Algorithm 1 for a given target representation $i$ from layer $(\ell)$. At each round $t$, the update algorithm maintains a probability vector $\pi_t$ over a set of all possible actions from routing choice space $\mathcal{P}$. The algorithm chooses a routing choice $a_t = (j_t \to \ell, \bigoplus^t)$ randomly drawn according to the probability vector $\pi_t$ (in Line 7). Here $j_t$ is the selected source representation to be transfered to the target layer $l$ and combined with target representation $i$ using the operator $\bigoplus^t$.

*Reward function:* The reward $r_t$ for the selected routing choice is then computed by evaluating gain in the loss due to the chosen source-target combination as follows: the prediction gain is the difference between the target network's losses on a hold out set $D_v$ with and without the routing choice $a_t$ i.e., $\mathcal{L}(f_T^M(x)) - \mathcal{L}(\tilde{f}_T^M(x))$ for a given image $x$ from the hold out data. This is shown in the Algorithm 3 EVALUATE. The reward function is used in Lines 4 and 5 to update the probability vector $\pi_{p,t}$ almost identical to the update in the classical EXP3.P algorithm of (Auer et al., 2002). Note that if the current version of the trainable parameters is not available, then a random initialization is used. In our experiments, this reward value is mapped to the $[-1, 1]$ range to feed as a reward to the bandit update algorithm.

*Environment Update:* Given the choice $j \to i$ and the operator $\bigoplus$, the target network is trained for one epoch over all samples in the training data $D_T$ for the target task. Algorithm 2 TRAIN-TARGET updates the target network weights $\mathcal{W}_T$ and other trainable parameters $(\mathcal{W}_{i,j}^{\bigoplus}, T_j)$ of the routing choice $a_t$ for each epoch on the entire target training dataset. Our main goal is to train the best target network that can effectively combine the best source representation chosen. Here, $\mathcal{L}$ is the loss function which operates on the final representation layer of the target network. $\alpha_t = 1/t$ and $\beta$ is the exploration parameter. We set $\beta = 0.4$ and $\gamma = 10^{-3}$.

---

**Algorithm 1** AMAB - Update Algorithm for Target Layer $\ell$

---

1: **Inputs:** Learning rate $\alpha_t$, Exploration parameter $\beta$, Number of Epochs $E$. Routing choice set $\mathcal{P}$
   **Initialize:** $w_{0,p}, \tilde{r}_{0,p} \leftarrow 0$.
2: **for** $t \in [1:E]$ **do**
3:   **for** $p \in \mathcal{P}$ **do**
4:     $w_{t,p} \leftarrow \log\left[(1-\alpha_t)\exp\left\{w_{t-1,p}+\gamma\tilde{r}_{t-1,p}\right\} + \frac{\alpha_t}{K-1}\sum_{j\neq p}\exp\left\{w_{t-1,j}+\gamma\tilde{r}_{t-1,j}\right\}\right]$
5:

$$\pi_{t,p} \leftarrow (1-\beta)\frac{e^{w_{t,p}}}{\sum_{j=1}^{K}e^{w_{t,j}}} + \frac{\beta}{K} \tag{2}$$

6:   **end for**
7:   Choose action $a_t \sim \pi_t$. Let $a_t = (j_t \rightarrow \ell, \bigoplus^t)$.
8:   Obtain current version of trainable parameters: $\left(\mathcal{W}_T, T_{j_t}, \mathcal{W}_{i,j}^{\bigoplus^t}\right)$. Use the standard random initialization if not initialized.
9:   $r_{t,a_t} \leftarrow \text{EVALUATE}(a_t, \left(\mathcal{W}_T, T_{j_t}, \mathcal{W}_{i,j}^{\bigoplus^t}\right))$
10:   $\left(\mathcal{W}_T, T_{j_t}, \mathcal{W}_{i,j}^{\bigoplus^t}\right) \leftarrow \text{TRAIN-TARGET}(a_t, \left(\mathcal{W}_T, T_{j_t}, \mathcal{W}_{i,j}^{\bigoplus^t}\right))$
11:   $\tilde{r}_{t,p} \leftarrow \begin{cases} \frac{r_{t,p}}{\pi_{t,p}} & \text{if } p = a_t, \\ 0 & \text{otherwise} \end{cases}$
12: **end for**

---

**Algorithm 2** TRAIN-TARGET - Train Target Network

---

1: **Inputs:** Target training dataset $D_T$, Target loss $\mathcal{L}(\cdot)$. Routing choice: $(j \rightarrow i, \bigoplus)$. Seed weight parameters: $\mathcal{W}_T[0], T_j[0], \mathcal{W}_{i,j}^{\bigoplus}[0]$.
2: Randomly shuffle $D_T$.
3: **for** $k \in [1:|D_T|]$ **do**
4:   $x \leftarrow D_T[k]$.
5:   $\left(\mathcal{W}_T[k], T_j[k], \mathcal{W}_{i,j}^{\bigoplus}[k]\right) \leftarrow \left(\mathcal{W}_T[k-1], T_j[k-1], \mathcal{W}_{i,j}^{\bigoplus}[k-1]\right)$
$$-\eta_k \nabla_{\left(\mathcal{W}_T, T_j, \mathcal{W}_{i,j}^{\bigoplus}\right)} \mathcal{L}(\tilde{f}_T^M(x))$$
6: **end for**
7: **Output:** Last iterate of $\left(\mathcal{W}_T, T_j, \mathcal{W}_{i,j}^{\bigoplus}\right)$

---

**Algorithm 3** EVALUATE - Evaluate Target Network

---

1: **Inputs:** Routing Choice: $(j \rightarrow i, \bigoplus)$. Weight parameters: $\mathcal{W}_T, T_j, \mathcal{W}_{i,j}^{\bigoplus}$. Target Loss $\mathcal{L}()$. Target task hold out set $D_v$.
2: **Output:** $\frac{1}{|D_v|}\sum_{x\in D_v}\mathcal{L}(f_T^M(x)) - \mathcal{L}(\tilde{f}_T^M(x))$.

---

### 3.3 ROUTING CHOICES

The routing choice $(j \rightarrow i, \bigoplus_{i,j})$ can be seen as deciding *where, what and how* to transfer/combine the source representations with the target network.

*Where to transfer?* The routing function $j \rightarrow i$ decides which one of the $N$ intermediate source features is useful for a given target feature $f_T^i$. In addition to these combinations, we allow the routing function to ignore the transfer using the NULL option. This allows the target network to potentially discard the source knowledge if it's unrelated to the target task.

*What to transfer?* Once a pair of source-task $(j \rightarrow i)$ combination is selected, the routing function decides what relevant information from the source feature $f_S^j$ should be transferred to the target

network using the transformation $T_j$. We use a Convolution-BatchNorm block to transfer useful features to the target network $\tilde{f}_S^j = \mathrm{BN}(\mathrm{Conv}(f_S^j))$. Here, $T_j = \mathrm{BN}(\mathrm{Conv}(\cdot))$. The convolution layer can select for relevant channels from the source representation and the batch normalization (Ioffe & Szegedy, 2015) addresses the covariant-shift between the source and the target representations, we believe that this combination is sufficient to "match" the two representations. This step also ensures that the source feature has a similar shape to that of the target feature.

*How to transfer (i.e. combine the representations)?* Given a pair of source and target feature representations ($j \rightarrow i$), the routing function chooses one of the following operations (i.e. $\bigoplus$) to combine them. We describe the class of operations $\mathcal{M}$, i.e. the various ways (1) is implemented.

1. **Identity** (Iden) operation allows the target network just to use the target representation $f_T^i$ after looking at the processed source representation $\tilde{f}_S^j$ from the previous Conv-BN step.
2. **Simple Addition** (sAdd) adds the source and target features: $\tilde{f}_T^i = \tilde{f}_S^j + f_T^i$.
3. **Weighted Addition** (wAdd) modifies sAdd with weights for the source and target features. These weights constitute $\mathcal{W}_{i,j}^{\oplus}$. i.e. the trainable parameters of this operation choice: $\tilde{f}_T^i = w_{S,i,j} * \tilde{f}_S^j + w_{T,i,j} * f_T^i$.
4. **Linear Combination** (LinComb) uses the linear block (without bias term) along with the average pooling to weight the features: $f_T^i = \mathrm{Lin}_{S,i,j}(\tilde{f}_S^j) * \tilde{f}_S^j + \mathrm{Lin}_{T,i,j}(f_T^i) * f_T^i$ where $\mathrm{Lin}_{.,i,j}$ is a linear transformation with its own trainable parameters.
5. **Feature Matching** (FM) follows the earlier work and forces the target feature to be similar to the source feature. This operation adds a regularization term $w_{i,j}\|\tilde{f}_S^j - f_T^i\|$ to the target objective $\mathcal{L}$ when we train.
6. **Factorized Reduce** (FactRed) use two convolution modules to reduce the number of channels $c$ in the source and target features to $c/2$ and concat them together: $f_T^i = \mathrm{concat}(\mathrm{Conv}_{S,i,j}^{c/2}(\tilde{f}_S^j), \mathrm{Conv}_{T,i,j}^{c/2}(f_T^i))$.

An action $a$ from the search space is given by $[(j \rightarrow i), \bigoplus_{i,j}]$. The total number of choice combinations is $\mathcal{O}((N + 1)M)$. Typically $N$ and $M$ are very small numbers, for instance, when Resnet is used as a source and target networks, we have $N = 4, M = 5$. For large action search spaces, action pruning (Even-Dar et al., 2006) and greedy approaches (Bayati et al., 2020) can be used to efficiently learn the best combinations as demonstrated in our experiment section.

## 4 EXPERIMENTS

In this section, we present experimental results to validate our Auto-Transfer methods. We first show the improvements in model accuracy that can be achieved over various baselines on six different datasets (section A.3) and two network/task setups. We then demonstrate superiority in limited sample size and limited training time usecases. Finally, we use visual explanations to offer insight as to why performance is improved using our transfer method. Experimental results on a toy example can be found in the supplement section A.1.

### 4.1 EXPERIMENTAL SETUP

Our transfer learning method is compared against existing baselines on two network/task setups. In the first setup, we transfer between similar architectures of different complexities; we use a 34-layer ResNet (He et al., 2016) as the source network pre-trained on ImageNet and an 18-layer ResNet as the target network. In the second setup, we transfer between two very different architectures; we use an 32-layer ResNet as the source network pretrained on TinyImageNet and a 9-layer VGG (Simonyan & Zisserman, 2014) as the target network. For ImageNet based transfer, we apply our method to four target tasks: Caltech-UCSD Bird 200 (Wah et al., 2011), MIT Indoor Scene Recognition (Quattoni & Torralba, 2009), Stanford 40 Actions (Yao et al., 2011) and Stanford Dogs (Khosla et al., 2011). For TinyImageNet based transfer, we apply our method on two target tasks: CIFAR100 (Krizhevsky et al., 2009), STL-10 (Coates et al., 2011).

We investigate different configurations of transfer between source and target networks. In the *full* configuration, an adverserial multi-armed bandit (AMAB) based on Exponential-weight algorithm

for Exploration and Exploitation (EXP3) selects (source, target) layer pairs as well as one of one of five aggregation operations to apply to each pair (operations are independently selected for each pair). In the *route* configuration, the AMAB selects layer pairs but the aggregation operation is fixed to be weighted addition. In the *fixed* configuration, transfer is done between manually selected pairs of source and target layers. Transfer can go between any layers, but the key is that the pairs are manually selected. In each case, during training, the source network is passive and only shares the intermediate feature representation of input images hooked after each residual block. After pairs are decided, the target network does aggregation of each pair of source-target representation in feed-forward fashion. The weight parameters of aggregation are trained to act as a proxy to how much source representation is useful for the target network/task. For aggregating features of different spatial sizes, we simply use a bilinear interpolation.

## 4.2 Experiments on Transfer Between Similar and Different Architectures

In the first setup, we evaluate all three Auto-Transfer configurations, full, fixed, and route, on various visual classification tasks, where transfer is from a Resenet-34 model to a Resnet-18 model. Our findings are compared with an independently trained Resnet-18 model (Scratch), another Resnet-18 model tuned for ImageNet and finetuned to respective tasks (Finetune), and the following existing baselines: Learning without forgetting (LwF) (Li & Hoiem, 2017), Attention Transfer (AT) (Zagoruyko & Komodakis, 2016), Feature Matching (FM) (Romero et al., 2014), Learning What and Where to Transfer (L2T-ww) (Jang et al., 2019) and Show, Attend and Distill (SAaD) (Ji et al., 2021). Results are shown in Table 6. Each experiment is repeated 3 times.

First, note that the Auto-Transfer Fixed configuration already improves performance on (almost) all tasks as compared to existing benchmarks. The fixed approach lets the target model decide how much source information is relevant when aggregating the representations. This result supports our approach to feature combination and demonstrates that it is more effective than feature matching. This even applies to the benchmark methods that go beyond and learn where to transfer to. Next, note that the Auto-Transfer Route configuration further improves the performance over the one-to-one configuration across all tasks. For example, on the Stanford40 dataset, Auto-Transfer Route improves accuracy over the second best baseline by more than 15%. Instead of manually choosing source and target layer pairs, we automatically learn the best pairs through our AMAB setup (Table 5 shows example set of layers chosen by AMAB). This result suggests that learning the best pairs through our AMAB setup to pick source-target pairs is a useful strategy over manual selection as done in the one-to-one configuration. To further justify the use of AMAB in our training, we conducted an ablation experiment (section A.6) where we retrain Auto-Transfer (fixed) with bandit chosen layer pairs, and found that the results were sub-optimal.

Next, note that Auto-Transfer Full, which allows all aggregation operations, does well but does not outperform Auto-Transfer Route. Indeed, the Auto-Transfer Full results showed that selected operations were all leaning to weighted addition, but other operations were still used as well. We conjecture that weighted addition is best for aggregation, but the additional operations allowed in Auto-Transfer Full introduce noise and make it harder to learn the best transfer procedure. Additionally, we conducted experiments by fixing aggregation to each of 5 operations and running Auto-Transfer Route and found that weighted addition gave best performance Table 8.

In order to demonstrate that our transfer method does not rely on the source and target networks being similar architectures, we proceed to transfer knowledge from a Resnet-32 model to a VGG-9 model. Indeed, Table 6 in the appendix demonstrates that Auto-Transfer significantly improves over other baselines for CIFAR100 and STL-10 datasets. Finally, we conducted experiments on matched configurations, where both Auto-Transfer (Route) and FineTune used same sized source and target models and found that Auto-Transfer outperforms FineTune (Figure 7 and Table 3).

## 4.3 Experiments on limited amounts of training samples

Transfer learning emerged as an effective method due to performance improvements on tasks with limited labelled training data. To evaluate our Auto-Transfer method in such data constrained scenario, we train our Auto-Transfer Route method on all datasets by limiting the number of training samples. We vary the samples per class from 10% to 100% at 10% intervals. At 100%, Stanford40 has ∼100 images per class. We compare the performance of our model against Scratch and L2T-ww

Table 1: *Transfer between Resnet models:* Classification accuracy (%) of transfer learning from ImageNet ($224 \times 224$) to Caltech-UCSD Bird 200 (CUB200), Stanford Dogs datasets, MIT Indoor Scene Recognition (MIT67) and Stanford 40 Actions (Stanford40). ResNet34 and ResNet18 are used as source and target networks respectively. Best results are bolded and each experiment is repeated 3 times. *DNR: did not report

| Source task | ImageNet | | | |
|---|---|---|---|---|
| Target task | CUB200 | Stanford Dogs | MIT67 | Stanford40 |
| Scratch | $39.11 \pm_{0.52}$ | $57.87 \pm_{0.64}$ | $48.30 \pm_{1.01}$ | $37.42 \pm_{0.55}$ |
| Finetune | $41.38 \pm_{2.96}$ | $54.76 \pm_{3.56}$ | $48.50 \pm_{1.42}$ | $37.15 \pm_{3.26}$ |
| LwF | $45.52 \pm_{0.66}$ | $66.33 \pm_{0.45}$ | $53.73 \pm_{2.14}$ | $39.73 \pm_{1.63}$ |
| AT | $57.74 \pm_{1.17}$ | $69.70 \pm_{0.08}$ | $59.18 \pm_{1.57}$ | $59.29 \pm_{0.91}$ |
| LwF+AT | $58.90 \pm_{1.32}$ | $72.67 \pm_{0.26}$ | $61.42 \pm_{1.68}$ | $60.20 \pm_{1.34}$ |
| FM | $48.93 \pm_{0.40}$ | $67.26 \pm_{0.88}$ | $54.88 \pm_{1.24}$ | $44.50 \pm_{0.96}$ |
| L2T-ww | $65.05 \pm_{1.19}$ | $78.08 \pm_{0.96}$ | $64.85 \pm_{2.75}$ | $63.08 \pm_{0.88}$ |
| SAaD | $68.29 \pm_{DNR}$ | $76.06 \pm_{DNR}$ | $66.47 \pm_{DNR}$ | $67.92 \pm_{DNR}$ |
| Auto-Transfer | | | | |
| - full | $67.86 \pm_{0.70}$ | $84.07 \pm_{0.42}$ | $74.79 \pm_{0.60}$ | $77.40 \pm_{0.74}$ |
| - fixed | $64.86 \pm_{0.06}$ | $86.10 \pm_{0.08}$ | $69.44 \pm_{0.41}$ | $77.27 \pm_{0.32}$ |
| - route | $\mathbf{74.76} \pm_{\mathbf{0.39}}$ | $\mathbf{86.16} \pm_{\mathbf{0.24}}$ | $\mathbf{75.86} \pm_{\mathbf{1.01}}$ | $\mathbf{80.10} \pm_{\mathbf{0.58}}$ |

for Stanford40 and report results in Figure 2 (top). Auto-Transfer Route significantly improves the performance over existing baselines. For example, at 60% training set (∼60 images per class), our method achieves 77.90% whereas Scratch and L2T-ww achieve 29% and 46%, respectively. To put this in perspective, Auto-Transfer Route requires only 10% images per class to achieve better accuracy than achieved by L2T-ww with 100% of the images. We see similar performance with other three datasets: CUB200, MIT67, Stanford Dogs (Figure 9).

## 4.4 IMPROVEMENTS IN TRAINING & INFERENCE TIMES

In order to assess training metrics and stability of learning, we visualize the test accuracy over training steps in Figure 2 (bottom) for the Stanford40 dataset. The results show that our method learns significantly quicker relative to the second closest baseline. For example, at epoch 25, our method achieves 74.55% accuracy whereas L2T-ww and Scratch achieve 25.55% and 21.69%, respectively. In terms of training time, Auto-Transfer Route took ∼300 minutes to train 200 epochs on the Stanford40 dataset, whereas L2T-ww and Scratch models took 610 and 170 minutes, respectively. Taken together, our method significantly improves performance over the second baseline with less than half the runtime. We report additional experiments with training curves plotted against training time in appendix (Figure 7) and inference times plotted against test accuracy (Figure 8). In Table 4 we show that for inference time matched models, Auto-Transfer (Route) outperforms FineTune by significant margin.

## 4.5 VISUAL EXPLANATIONS

In order to qualitatively analyze what bandit Auto-Transfer Route is learning, Grad-CAM (Selvaraju et al., 2017) based visual explanations are presented in Figure 3

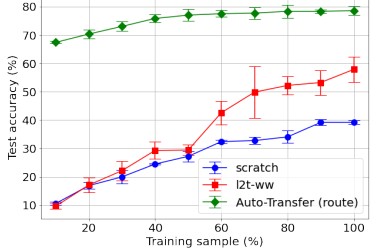

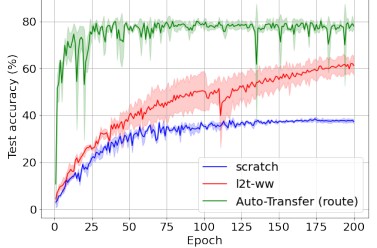

Figure 2: Above we see test accuracies as a function of (target) training sample size (top) and number of epochs (bottom) on the Stanford40 dataset for the Scratch model, L2T-ww (our closest competitor) and our method Auto-Transfer. Qualitatively similar behavior is also seen on the other datasets. Experiments repeated 3 times.

(additional explanations are in Figures 11, 12, and 13 in the appendix). Grad-CAM highlights pixels that played an important role in correctly labelling the input image. For each target task, we present a (random) example image that is correctly labelled by bandit Auto-Transfer but incorrectly classi-

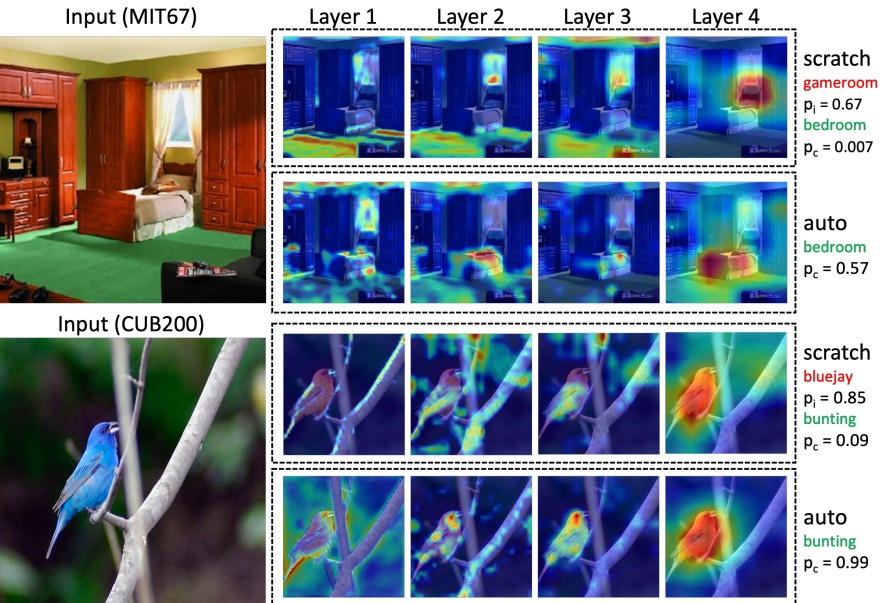

Figure 3: Layer-wise Grad-CAM images highlighting important pixels that correspond to predicted output class. We show examples from MIT67 and CUB200 (ImageNet based transfer) where the independently trained scratch model predicted the input image incorrectly, but our bandit based auto-transfer method predicted the right class for that image. Correctly predicted class is indicated in green text and incorrectly classified class is indicated in red text. Class probability for these predictions is also provided.

fied by Scratch, along with layer-wise Grad-CAM images that illustrate what each layer of the target model focuses on. For each image, we report the incorrect label, correct label and class probability for correct ($p_c$) and incorrect ($p_i$) labels.

Overall, we observe that our method pays attention to relevant visual features in making correct decisions. For example, in the first image from MIT67 dataset, the Scratch model incorrectly labelled it as a gameroom while the correct class is bedroom ($p_i = 0.67$, $p_c = 0.007$). The Grad-CAM explanations show that layers 1-3 of the Scratch model pay attention to the green floor which is atypical to a bedroom and common in gamerooms (e.g. pool tables are typically green). The last layer focuses on the surface below the window that looks like a monitor/tv that is typically found in gamerooms. On the other hand, our model correctly identifies the class as bedroom ($p_c = 0.57$) by paying attention to the bed and surrounding area at each layer.

To visualize an example from a harder task, consider the indigo bunting image from the CUBS dataset. The Scratch model classifies the image as a bluejay ($p_i = 0.85, p_c = 0.09$), but our model correctly predicts it as a bunting ($p_c = 0.99$). Indigo buntings and blue jays are strikingly similar, but blue jays have white faces and buntings have blue faces. We clearly see this attribute picked up by the bandit Auto-Transfer model in layers 2 and 3. We hypothesize that the source model, trained on millions of images, provides useful fine-grained information useful for classifying similar classes.

## 5   CONCLUSION

In this paper, we have put forth a novel perspective where we leverage and adapt an adversarial multi-armed bandit approach to transfer knowledge across heterogeneous tasks and architectures. Rather than constraining target representations to be close to the source, we dynamically route source representations to appropriate target representations also combining them in novel and meaningful ways. Our best combination strategy of weighted addition leads to significant improvement over state-of-the-art approaches on four benchmark datasets. We also observe that we produce accurate target models faster in terms of (training) sample size and number of epochs. Further visualization based qualitative analysis reveals that our method produces robust target models that focus on salient features of the input more so than its competitors, justifying our superior performance.

## ACKNOWLEDGMENT

We would like to thank Clemens Rosenbaum, Matthew Riemer, and Tim Klinger for their comments on an earlier version of this work. This work was supported by the Rensselaer-IBM AI Research Collaboration (`http://airc.rpi.edu`), part of the IBM AI Horizons Network (`http://ibm.biz/AIHorizons`).

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

## A    APPENDIX

### A.1    TOY EXAMPLE

In this section, we simulate our experiment on a toy example. We compare our Auto-Transfer with the other baselines: L2T-ww and Scratch. In this simulation, we consider Auto-Transfer with a fixed (one-to-one) setup for simplicity in our experiment analysis.

We consider predicting a *sine* wave function ($y = sin(x)$) as our source task and a *sinc* function ($y = \frac{sin(x)}{x}$) as our target task. Clearly, the features from the pretrained source model will help the target task in predicting the *sinc* function. Both the input data point $x$ and the output value $y$ are one-dimensional vectors ($d_{in} = d_{out} = 1$). We use a shallow linear network consists of 4 linear blocks: $f_1 = Lin_{(d_{in},h_1)}(x), f_2 = Lin_{(h_1,h_2)}(f_1), f_3 = Lin_{(h_1,h_2)}(f_2), out = Lin_{(h_3,d_{out})}(f_3)$ for a datapoint $x$. For source network, we set the hidden size to 64 (i.e., $h_1 = h_2 = h_3 = 64$) and 16 for the target network. We sampled $30,000$ data points to generate training set (x,y) and $10,000$ test-set data points for the source network and (i.e., $x$ is sampled from a Gaussian distribution and $y = sin(x)$). Similarly, we generated $1000$ training examples and $800$ test set examples for the target network. Both the source and the target networks are trained for $E = 50$ epochs.

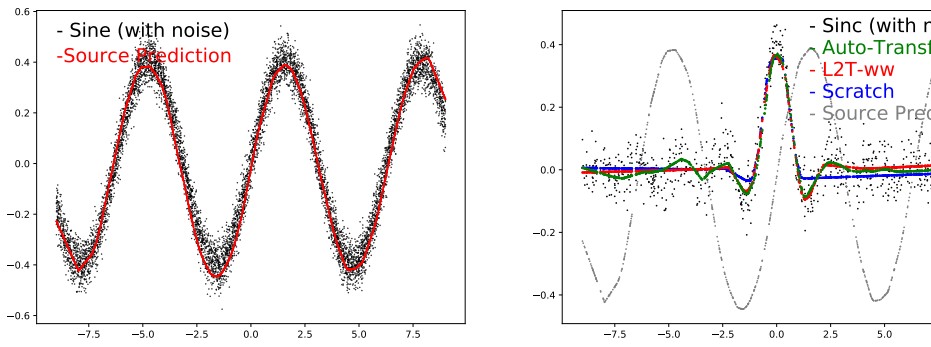

Figure 4: (Left) shows the test set data from the source task and the source models' prediction. (Right) shows the test-set predictions for target task data from Scratch, Source prediction, L2T-ww and Auto-Transfer with the shallow linear network configuration $[d_{in} = 1, h_1 = 16, h_2 = 16, h_3 = 16, d_{out} = 1]$.

Figure 4 (*left*) shows the source model prediction for the test data. Given the shallow linear network with 64 hidden dimensions and $30,000$ training example, the source model perfectly predicts the $sin(x)$ function. Figure 4 (*right*) shows the predictions from the scratch target model, source model, L2T-ww and Auto-Transfer for the target test data. We report the Auto-Transfer with fixed choice of [(0,0),(1,1),(2,2), wtAdd] for this experiment. We can see that the Auto-Transfer accurately predicts the target task even when there is a limited amount of labeled examples.

Our results show the test set loss for the target data is relatively less compared to the other baselines ($0.0030$ MSE loss for Auto-Transfer vs $0.0033$ and $0.125$ MSE loss for the scratch and L2T-ww). Figures 5 and 6 show The results on different network configurations and how the feature representations for Scratch, L2T-ww and Auto-Transfer changes over 50 training epochs.

### A.2    REAL DATASETS

We evaluate the performance of Auto-Transfer on six benchmarks with different tasks: Stanford Actions 40 dataset for action recognition, CUBS Birds 200 dataset for object recognition, Stanford Dogs 120 for fine-grained object recognition, MIT Indoors 67 for scene classification, CIFAR 100 and STL-10 image recognition datasets.

**Stanford Actions 40**. Stanford Actions 40 dataset contains images of humans performing 40 actions. There are about 180-300 images per class. We do not use bounding box and other annotation information for training. There are a total of 9,532 images, making it the smallest dataset in our benchmark experiments.

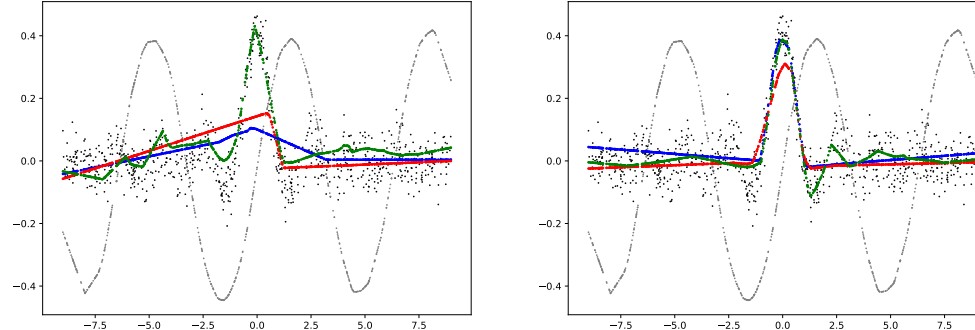

Figure 5: Test-set predictions for Scratch, Source prediction, L2T-ww and Auto-Transfer for the target task data with the shallow linear network configurations *left*: $[d_{in} = 1, h_1 = 4, h_2 = 4, h_3 = 4, d_{out} = 1]$, *right*: $[d_{in} = 1, h_1 = 8, h_2 = 8, h_3 = 8, d_{out} = 1]$

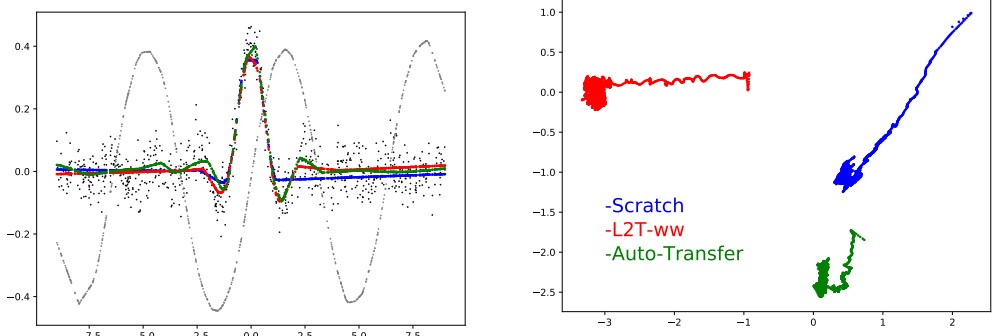

Figure 6: Test-set predictions for Scratch, Source prediction, L2T-ww and Auto-Transfer for the target task data with different choices from the routing function *left*: [(2,0),(-1,1),(-1,2), wAdd] *right*: show the feature representations of a single data point (plotted over the 50 training epochs) extracted from the final layer of the target network

**Caltech-UCSD Birds-200-2011**. CUB-200-2011 is a bird classification datset with 200 bird species. Each species is associated with a wikipedia article and organized by scientific classification. Each image is annotated with bounding box, part location, and attribute labels. We use only classification labels during training. There are a total of 11,788 images.

**Stanford Dogs 120**. The Stanford Dogs dataset contains images of 120 breeds of dogs from around the world. There are exactly 100 examples per category in the training set. It is used for the task of fine-grained image categorization. We do not use the bounding box annotations. There are a total of 20,580 images.

**MIT Indoors 67**. MIT Indoors 67 is a scene classification task containing 67 indoor scene categories, each of which consists of at most 80 images for training and 20 for testing. Indoor scene recognition is challenging because spatial properties, background information and object characters are expected to be extracted. There are 15,620 images in total.

**CIFAR 100**. CIFAR 100 is a image recognition task containing 100 different classes with 600 images in each class. There are 500 training images and 100 testing images per class. It is a subset of tiny images datastet.

**STL 10**. STL 10 is a image recognition task containing 10 classes. It is inspired by the CIFAR-10 dataset but with some modifications. In particular, each class has fewer labeled training examples than in CIFAR-10.

### A.3 EXPERIMENT DETAILS

For our experimental analysis in the main paper, we set the number of epochs for training to $E = 200$. The learning rate for SGD is set to $0.1$ with momentum $0.9$ and weight decay $0.001$. The learning rate for the ADAM is set to $0.001$ with and weight decay of $0.001$. We use Cosine Annealing learning rate scheduler for both optimizers. The batch size for training is set to $64$. Our target networks were randomly initialized before training.

The target models were trained in parallel on two machines with the specifications shown in Table 2.

| Resource | Setting |
|---|---|
| CPU | Intel(R) Xeon(R) CPU E5-2690 v4 @ 2.60GHz |
| Memory | 128GB |
| GPUs | 1 x NVIDIA Tesla V100 16 GB |
| Disk | 600GB |
| OS | Ubuntu 18.04-64 Minimal for VSI. |

Table 2: Resources used by Auto-Transfer

### A.4 TRAINING AND TESTING PERFORMANCE

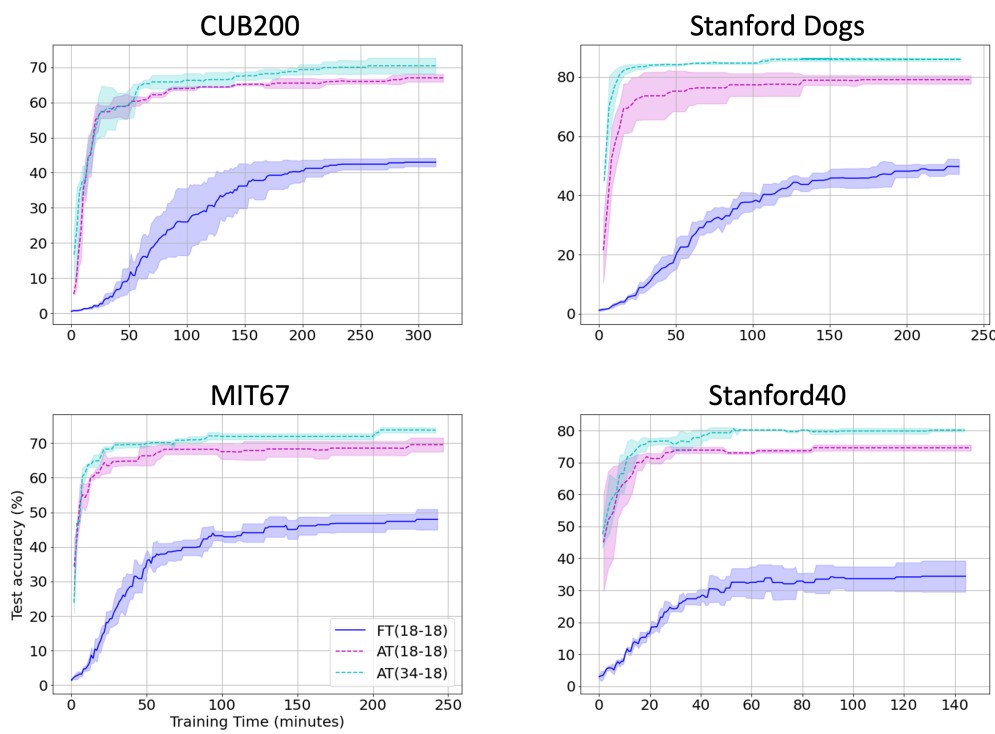

Figure 7: Above we see test accuracies as a function of training time (minutes) plotted for following architectures (i) Finetuning (ResNet18 - ResNet18), (ii) AutoTransfer (ResNet18 - ResNet18), (iii) AutoTransfer (ResNet34 - ResNet18), denoted FT(18-18), AT(18-18), and AT(34-18), respectively. We significantly outperform finetuning in all datasets.

### A.5 ADDITIONAL EXPERIMENTS ON LIMITED AMOUNTS OF DATA

To evaluate our Auto-Transfer method in data constrained scenario further, we train our Auto-Transfer (route) method on the CUB200, Stanford Dogs and MIT67 datasets by limiting the number of training samples (Figure 9). We vary the samples per class from 10% to 100% at 10% intervals.

Table 3: Classification accuracy (%) of transfer learning for matched architectures ResNet18 - ResNet18 for Auto-Transfer and Finetuning. Best results are bolded.

|  | CUB200 | Stanford Dogs | MIT67 | Stanford40 |
|---|---|---|---|---|
| Finetune (R18 - R18) | $42.96\pm_{1.45}$ | $53.02\pm_{3.57}$ | $47.93\pm_{3.66}$ | $34.40\pm_{5.94}$ |
| AutoTransfer (R18 - R18) | $\mathbf{66.97}\pm_{1.38}$ | $\mathbf{79.46}\pm_{1.05}$ | $\mathbf{69.54}\pm_{2.49}$ | $\mathbf{75.07}\pm_{2.55}$ |

Table 4: Average classification accuracy (%) and average inference times of transfer learning for time matched architectures using ResNet18 - ResNet18 for Auto-Transfer and ResNet34 - ResNet34 for Finetuning.

|  | CUB200 | | Stanford Dogs | | MIT67 | | Stanford40 | |
|---|---|---|---|---|---|---|---|---|
|  | t (sec) | % | t | % | t | % | t | % |
| Finetuning (R34 - R34) | 12.88 | 37.13 | 12.66 | 52.26 | 12.22 | 44.37 | 14.0 | 31.12 |
| Auto-Transfer (R18 - R18) | 14.46 | 64.37 | 13.83 | 77.07 | 14.26 | 67.89 | 15.28 | 69.02 |
| Auto-Transfer (R34 - R18) | 18.55 | 71.84 | 18.27 | 85.09 | 18.62 | 69.76 | 19.20 | 79.74 |

Table 5: Final source layer selected at 200th epoch for each target layer for 3 repetitions for Table 1 experiments.

|  | Selected source layer (run_1, run_2, run_3) | | | |
|---|---|---|---|---|
| Target Layer | Layer 1 | Layer 2 | Layer 3 | Layer 4 |
| CUB200 | 2, 2, 2 | 3, 2, 2 | 2, 1, 1 | 2, 4, 4 |
| Stanford Dogs | 1, 1, 4 | 3, 3, 5 | 2, 3, 2 | 4, 5, 4 |
| MIT67 | 2, 4, 2 | 3, 1, 5 | 2, 3, 1 | 3, 3, 4 |
| Stanford40 | 1, 2, 4 | 4, 3, 3 | 2, 3, 2 | 3, 4, 3 |

Table 6: *Transfer between Resnet model and VGG model:* Classification accuracy (%) of transfer learning from TinyImageNet to CIFAR100 and VGG9. ResNet32 and VGG9 are used as source and target networks respectively. Best results are bolded and each experiment is repeated 3 times.

| Source task | TinyImageNet | |
|---|---|---|
| Target task | CIFAR100 | STL-10 |
| Scratch | $67.69\pm_{0.22}$ | $65.18\pm_{0.91}$ |
| Finetune | $67.80\pm_{1.76}$ | $65.98\pm_{1.25}$ |
| LwF | $69.23\pm_{0.09}$ | $68.64\pm_{0.58}$ |
| AT | $67.54\pm_{0.40}$ | $74.19\pm_{0.22}$ |
| LwF+AT | $68.75\pm_{0.09}$ | $75.06\pm_{0.57}$ |
| FM | $69.97\pm_{0.24}$ | $76.38\pm_{0.88}$ |
| L2T-ww | $70.96\pm_{0.61}$ | $76.38\pm_{1.18}$ |
| Auto-Transfer | | |
| - full | $\mathbf{72.48}\pm_{0.42}$ | $78.46\pm_{1.10}$ |
| - fixed | $70.48\pm_{0.25}$ | $79.92\pm_{1.49}$ |
| - route | $70.89\pm_{0.36}$ | $\mathbf{82.09}\pm_{\mathbf{0.29}}$ |

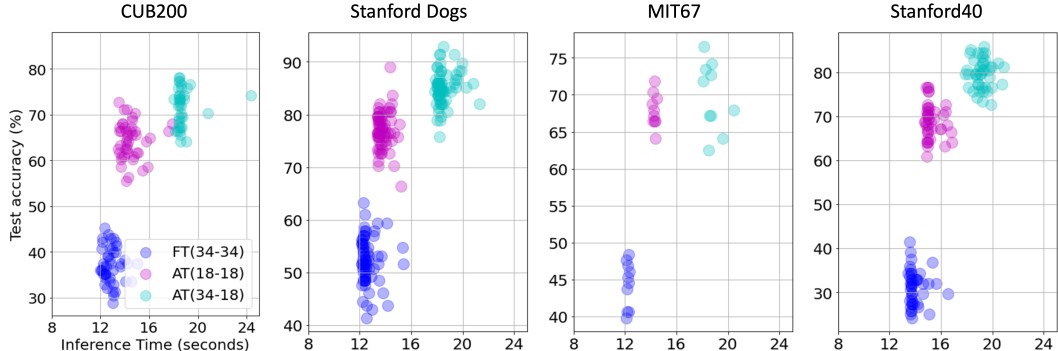

Figure 8: Test accuracies as a function of inference time plotted for following architectures (i) Finetuning (ResNet34 - ResNet34), (ii) AutoTransfer (ResNet18 - ResNet18), (iii) AutoTransfer (ResNet34 - ResNet18), denoted FT(34-34), AT(18-18), and AT(34-18), respectively. Each circle represents a batch of 128 sample images. We significantly outperform finetuning in all datasets.

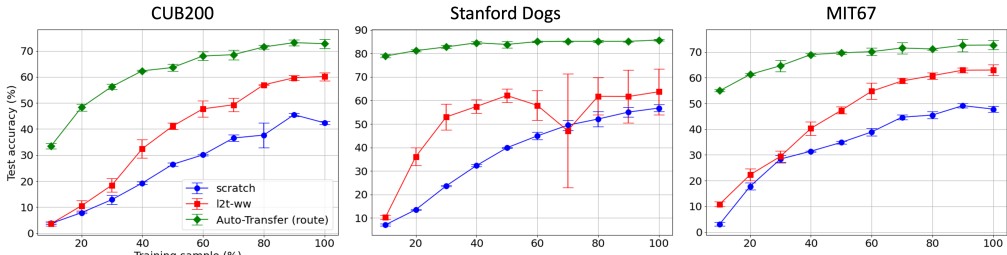

Figure 9: Above we see test accuracies as a function of (target) training sample size for CUB200, Stanford Dogs and MIT67 datasets. Each experiment is repeated 3 times.

## A.6 ABLATION STUDIES

### TRAINING THE NETWORK USING BANDIT SELECTED PAIRS

To evaluate the importance of training the target network with adversarial multi-armed bandit, we retrained our target network with a fixed source-layer configuration selected at 200th epoch of previous best bandit based experiments. For Eg. in our best bandit based experiment for CUB200, the source,target pairs were $\{(2,1), (3,2), (2,3), (2,4)\}$. As seen in Table 7, we find that this experiment decreased performance in comparison to bandit based one in all target tasks. This confirms the need for bandit based decision maker, that learns combination weights and pairs over training steps.

Table 7: Classification accuracy (%) of transfer learning ResNet34 to ResNet18 transfer where the source-target layer pairs are fixed to Auto-Transfer (route) selected ones at 200th epoch from previous runs.

| Task | CUB200 | Stanford Dogs | MIT67 | Stanford40 |
|---|---|---|---|---|
| Auto-Transfer (fixed, retrain) | 73.09 | 85.05 | 69.10 | 78.90 |
| Auto-Transfer (route) | **75.15** | **86.40** | **76.87** | **80.68** |

### TRAINING THE NETWORK USING DIFFERENT AGGREGATION OPERATORS

To evaluate how different aggregation operators influence Auto-Transfer, we train Auto-Transfer Route by fixing aggregation to 5 different operations. Identity (iden), Simple Addition (sAdd), Weighted Addition (wtAdd), Linear Combination (LinComb) and Factored Reduction (FactRed). Results for Stanford40 dataset is found in Table 8. We find that weighted addition performs the best.

Table 8: Classification accuracy (%) of transfer learning ResNet34 to ResNet18 transfer where the aggregation operator is fixed to Identity (iden), Simple Addition (sAdd), Weighted Addition (wtAdd), Linear Combination (LinComb) and Factored Reduction (FactRed).

|  | Iden | SAdd | WtAdd | LinComb | FactRed |
|---|---|---|---|---|---|
| Auto-Transfer (route) | 37.56 | 77.78 | **80.10** | 76.6 | 76.66 |

## A.7 VISUALIZING INTERMEDIATE REPRESENTATIONS

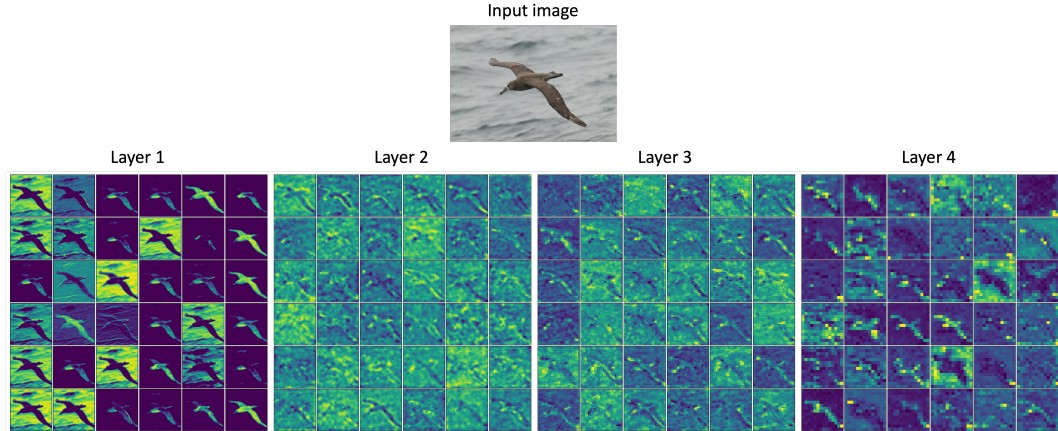

Figure 10: Example of learned intermediate representations for a bird image from CUB200 dataset. We plot the first 36 features in each layer ( there are 64, 128, 256 and 512 features for layers 1 to 4). It is hard to draw meaningful patterns by looking at intermediate representations, and hence we chose to investigate layer-wise Grad-CAM images.

## A.8 ADDITIONAL EXPLANATIONS USING GRAD-CAM

We here offer more examples of visual explanations of what is being transferred using Auto-Transfer Route. The first example in Figure 11 is an image of cooking from the Stanford40 dataset. The Scratch model incorrectly classifies the image as cutting ($p_i = 0.88, p_c = 0.01$) by paying attention to only the cooking surface that looks like a table and person sitting down (typical for someone cutting vegetables). On the other hand, our model correctly labels the image ($p_c = 0.99$) by paying attention to the wok and cooking utensils such as water pot, etc. We hypothesize that this surrounding information is provided by the source model which is useful in making the correct decision.

The second example in Figure 11 is from the Stanford Dogs dataset (Figure 11). The scratch model fails to pay attention to relevant class information (dog) and labels a chihuahua as german sheperd ($p_i = 0.23, p_c = 0.0002$) by focusing on the flower, while our method picks the correct label ($p_c = 0.99$). Bandid Auto-Transfer gets knowledge about the flower early on and then disregards this knowledge before attending to relevant class information. Further examples of visual explanations comparing to L2T-ww (Figure 12) and counter-examples where our method identifies the wrong label (Figure 13) follow below. For these counter-examples we find that the task is typically hard. For eg. playing violin vs playing guitar. And, the class probability of incorrect label is closer to that of correct label, suggesting that our method was not confident in predicting wrong class.

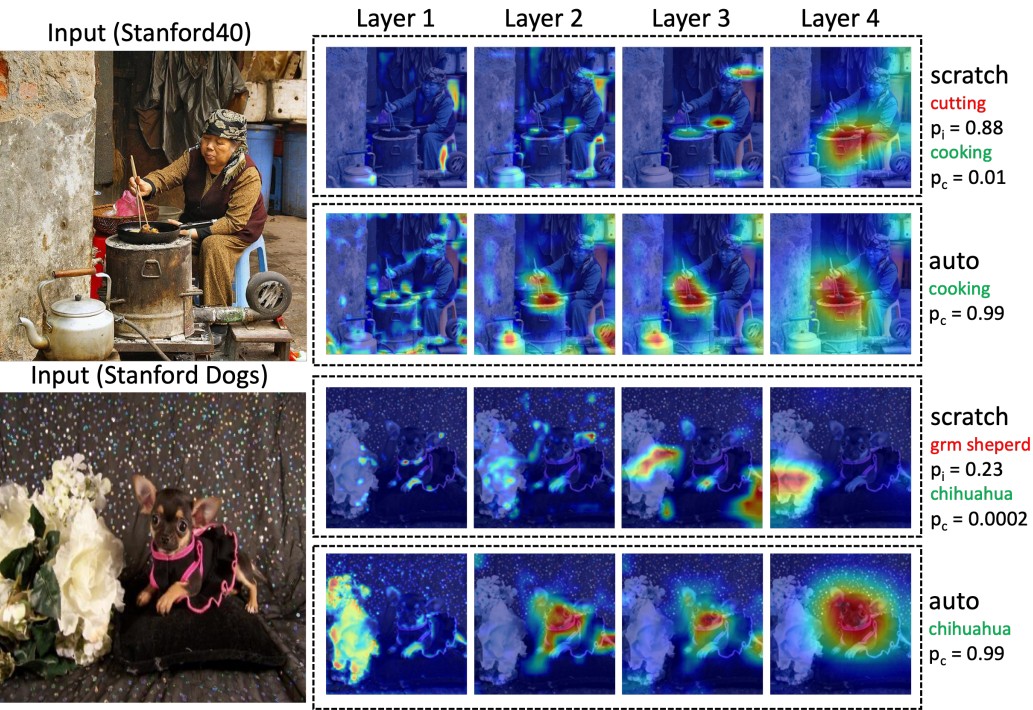

Figure 11: Layer-wise Grad-CAM images highlighting important pixels that correspond to predicted output class. We show examples from Stanford40 and Stanford Dogs (ImageNet based transfer) where the independently trained scratch model predicted the input image incorrectly, but our bandit based auto-transfer method predicted the right class for that image. Correctly predicted class is indicated in green text and incorrectly classified class is indicated in red text. Class probability for these predictions is also provided.

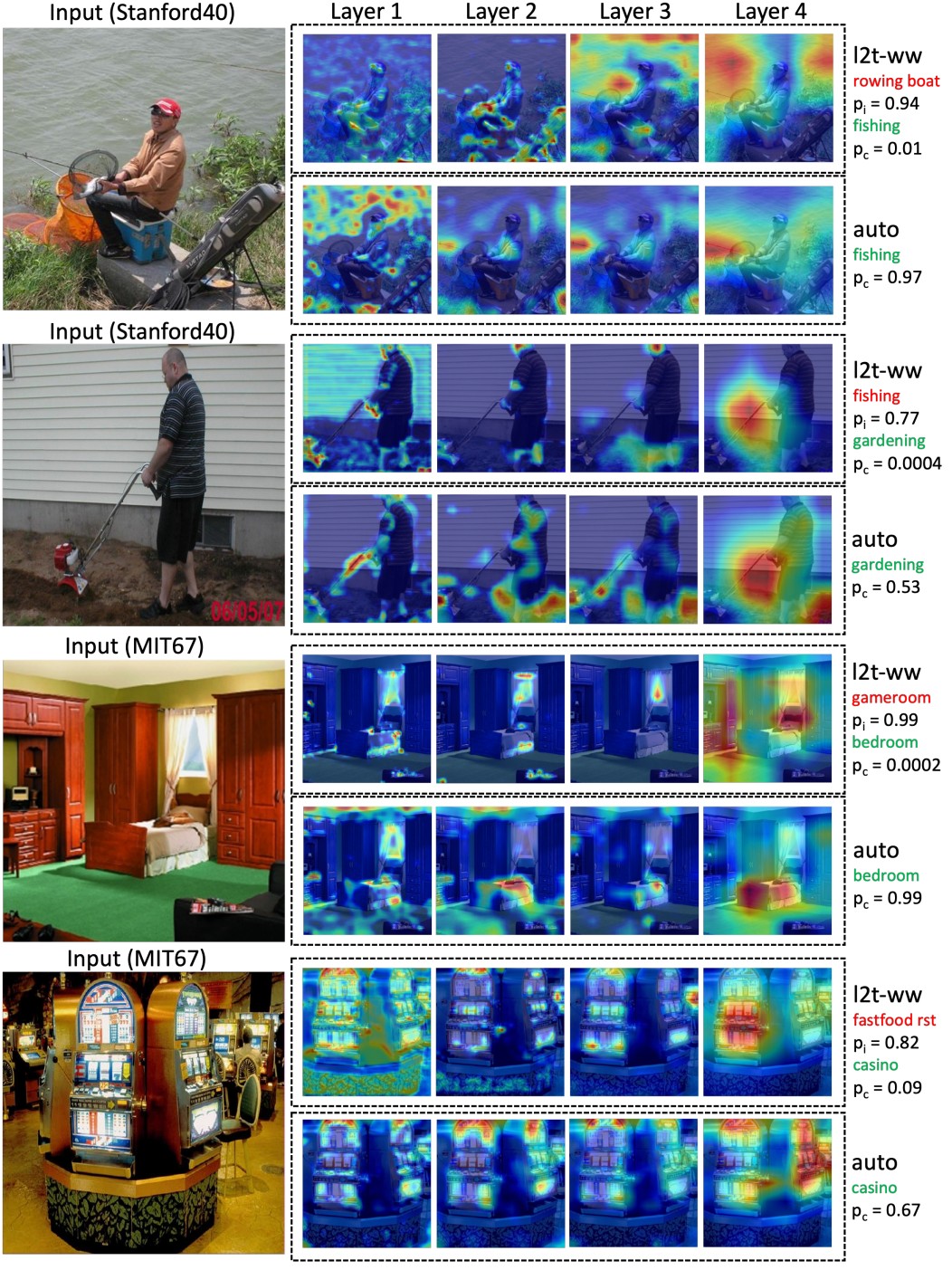

Figure 12: Layer-wise Grad-CAM images highlighting important pixels that correspond to predicted output class. We show examples where the L2T-ww model predicted the input image incorrectly, but our bandit based auto-transfer method predicted the right class for that image. Correctly predicted class is indicated in green text and incorrectly classified class is indicated in red text. Class probability for these predictions is also provided.

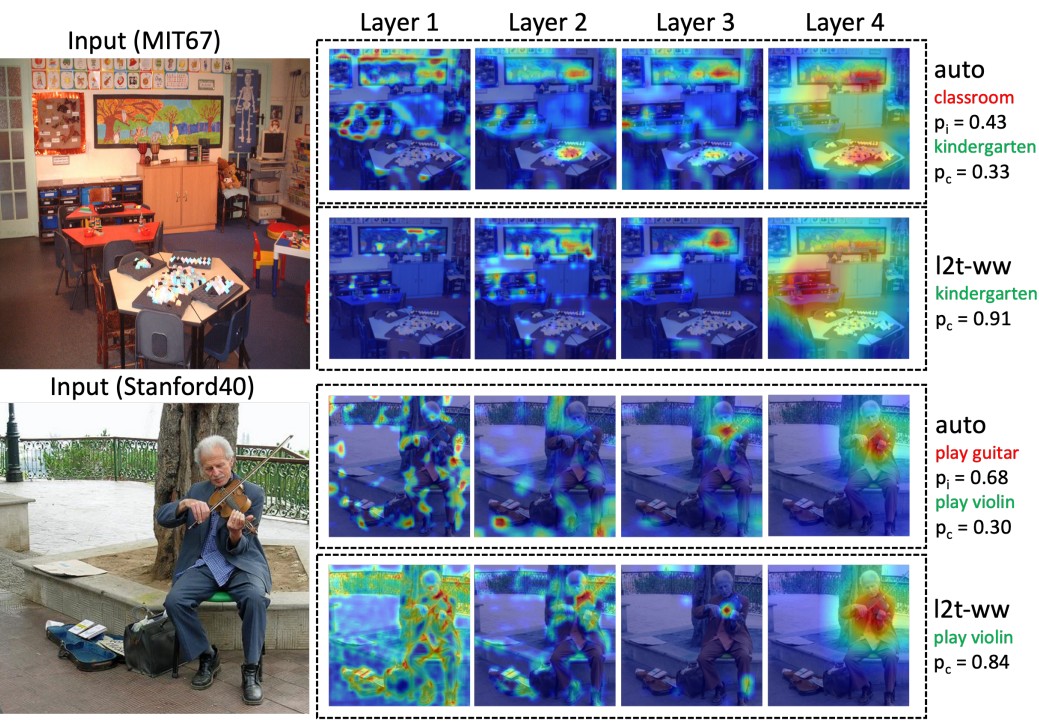

Figure 13: Layer-wise Grad-CAM images highlighting important pixels that correspond to predicted output class. We show examples where the L2T-ww model predicted the input image correctly, but our bandit based auto-transfer method predicted the wrong class for that image. Correctly predicted class is indicated in green text and incorrectly classified class is indicated in red text. Class probability for these predictions is also provided.

