# OpenReview forum: "Auto-Transfer: Learning to Route Transferable Representations"
_ICLR.cc/2022/Conference — ICLR 2022 Poster_

### Official Review · Reviewer_8d9w · 2021-10-30

**Correctness:** 2
**Technical Novelty And Significance:** 3
**Empirical Novelty And Significance:** 3
**Recommendation:** 6
**Confidence:** 5

**Details Of Ethics Concerns:**

As described by the authors, it is important to be alert about the possible biases transferred from the source domain/model into future target models. This is probably true for every single domain transfer method. I believe this is not a blocker for publication, and authors clearly reflect this in their manuscripts, still, it is good to keep it in mind.

**Main Review:**

Pros:

- The proposed approach does a good job at transferring representations between heterogeneous CNNs, showing good results in several image classification datasets and improving upon state-of-the-art solutions.

- The algorithm is relatively easy to implement and results should be reproducible. Code is provided.

##########################################################################
Cons:


- There seems to be a disconnect between the general idea presented in the paper and the “flavor” of the approach that yields the best results. In practice, the idea of using multi-arm bandits to learn features mapping along with a set of transformation and combination operators seems to be reduced to learning the feature mapping fixing those operators. If this is the case, the narrative of the paper should reflect the finding (from the beginning of the manuscript). If this is not the case, empirical evidence should support it.
This observation affects the consistency of the paper and triggers multiple questions that need to be addressed during the rebuttal period.

##########################################################################
Questions during rebuttal period:
Why did authors choose a normalized linear operator for the transformation of the source features (i.e., Tj = BN(Conv(·)) )? Were other operators analyzed and discarded? Is the idea just to adjust channel dimensions and address the covariant shift?

From table 1, it seems that fixing the combination operator to be a weighted addition and the aggregator operator to be a bilinear interpolation yields the best results. The “full” configuration, in which the multi-arm bandits’ algorithm has to select these operators comes in second place. Shouldn’t this observation deeply affect the original hypothesis of this work as well as the global structure of the paper?

As a follow-up to the previous question, could the authors provide additional details on the “fixed configuration”? From the text, it is clear that the feature maps are done manually, but is the combination operator set to a weighted addition, and is the aggregator operator set to a bilinear interpolation? After observing how the
“route configuration” outperforms the “full configuration” I wonder if the “fixed configuration” could be further improved by fixing those operators to a weighted addition and a bilinear interpolation. Could authors please report a configuration with these details?

**Summary Of The Paper:**

The paper presents a new approach to perform transfer learning between heterogeneous architectures and tasks by introducing a new algorithm based on adversarial multi-armed bandit, which automatically learns a mapping between source and target representations, as well as the way to combine such representations. The approach is evaluated in multiple vision tasks, starting from models pre-trained on ImageNet, producing satisfactory results when compared to the state-of-the-art.

**Summary Of The Review:**

I am currently indecisive, leaning towards rejection. The paper presents a nice narrative about the adversarial multi-armed bandit approach, highlighting the benefits of the routing of representations in combination with the dynamic selection of operators to combine and transform features. However, experiments seem to hint at a different reality, where fixing these operators is indeed more beneficial.  Based on this, I consider that authors need to present stronger evidence that supports the value of the approach, as it is currently presented in the paper, or they need to update the narrative of the paper to account for the empirical findings.

Please, consider my questions for rebuttal.

Post-rebuttal review:

After the evidence provided by the authors, I have decided to increase the valuation to a 6: marginally accepted. The new tables are able to defend the value of the main idea, but still, there are some doubts about the potential of "full" vs. "route" in different problems/datasets. This is the main reason for not assigning a higher score.

---

> ### Author Response · Authors · 2021-11-18
> **Response to Reviewer 8d9w**
>
> We thank reviewer 8d9w for helping improve our work. We address the concerns below.
>
> **Clarification about bandits**
>
> #### Q1. Why did authors choose a normalized linear operator for the transformation of the source features (i.e., Tj = BN(Conv(·)) )? Were other operators analyzed and discarded? Is the idea just to adjust channel dimensions and address the covariant shift?
>
> A1. The reviewer is correct. We wanted to preserve all the information available in the source features and let the aggregation operator take care of how to combine source features with target features.
>
> #### Q2. From table 1, it seems that fixing the combination operator to be a weighted addition and the aggregator operator to be a bilinear interpolation yields the best results. The “full” configuration, in which the multi-arm bandits’ algorithm has to select these operators comes in second place. Shouldn’t this observation deeply affect the original hypothesis of this work as well as the global structure of the paper?
>
> A2. We apologize for the confusion. The search space of adversarial multi-armed bandit (AMAB) includes source - target layer pairs and aggregation operator,$[(j \rightarrow i),  \bigoplus_{i,j}]$. There is no separate combination operation. After training the Auto-Transfer(full) configuration using this search space on several target tasks, we found that the aggregation operation tends to be $\bigoplus_{i,j}= wtAdd$. To test this further, we fixed the aggregation operation and trained the network using Auto-Transfer (route), where the search space for AMAB is reduced to $[(j \rightarrow i)]$. We report the results from Stanford40 dataset below and in Table 8,
>
> |                       | Iden              | SAdd              | WtAdd             | LinComb           | FactRed           |
> |-----------------------|-------------------|-------------------|-------------------|-------------------|-------------------|
> | Auto-Transfer (route) | 37.56&pm;1.31 | 77.78&pm;1.18 | **80.11**&pm;0.58 | 76.60&pm;2.50 | 76.66&pm;2.70 |
>
> As seen from above, fixing aggregation to wtAdd resulted in best performance. We would like to point that AMAB is still deciding which layer pairs to combine and wtAdd operation decides how much of source and target representations to combine using source and target weight parameters: $\tilde f_{T}^i = w_{S,i,j} * \tilde f_{S}^j + w_{T,i,j} * f_{T}^i$.
>
> Please see our response A3 to reviewer 7soo for more details on this, where we also highlight the fact that in our experiments on additional datasets (Table 3) wtAdd was not consistently best performing thus further motivating the need for the Auto-Transfer (full) approach.
>
> **Clarification about configurations**
>
> #### Q3. As a follow-up to the previous question, could the authors provide additional details on the “fixed configuration”?... After observing how the “route configuration” outperforms the “full configuration” I wonder if the “fixed configuration” could be further improved by fixing those operators to a weighted addition and a bilinear interpolation. Could authors please report a configuration with these details?
>
> A3. Please find below our explanation for the different configurations of Auto-Transfer reported in Table 1,
>
> **Auto-Transfer (full)** decides source - target layer pairs and aggregation operation: $[(j \rightarrow i),  \bigoplus_{i,j}]$
>
> - ex: [(2 → 1, sAdd), (3 → 2, lincomb), (1 → 3, wtAdd), (None → 4, wtAdd)]
>
> **Auto-Transfer (route)** decides source-target layer pairs while fixing aggregation to weighted addition: $[(j \rightarrow i), \bigoplus_{i,j}= wtAdd]$
> - ex: [(2 → 1, wtAdd), (3 → 2, wtAdd), (1 → 3, wtAdd), (None → 4, wtAdd)]
>
> **Auto-Transfer (fixed)** uses fixed source-target layer pairs and weighted addition:
> - ex: [(1 → 1,wtAdd),(2 → 2,wtAdd),(3 → 3,wtAdd),(4 → 4,wtAdd)]
>
> As detailed above and A2, there is no separate combination and aggregation operations. Auto-Transfer (fixed) is the only *fixed configuration*. Please note that even in this case, the network has the liberty to choose the amount of source and target representations to combine since a scalar weight is learned. We hope that this clarifies the concerns raised. In addition, we have added these clarifications in the revised version.

---

> > ### Comment · Reviewer_8d9w · 2021-11-25
> > **Still concerned about results**
> >
> > I would like to thank the authors for their clarification.
> >
> > Some of my questions still apply. You mention that Table 3 shows the value of Auto-transfer (full). If that is the case, that table should be part of the main paper. Also, is the entry labeled as "AutoTransfer (R18 - R18)" the "full", the "route", or the "fixed" version? Where are the other entries?

---

> > > ### Author Response · Authors · 2021-11-25
> > > **Correction on our previous response: (Table 6 instead of Table 3)**
> > >
> > > We greatly appreciate your continued discussion. We must apologize - we meant to write Table 6 instead of Table 3. Table 6 shows Auto-Transfer full vs fixed vs route as you expected. This table was meant simply to offer additional datasets for comparison to make the paper more complete as requested by Reviewer 3TiT.  As noted, the value of Auto-transfer(full) is illustrated here on the transfer to CIFAR100. In a final version, we are happy to append this table to Table 1 in the main paper (which we note will not take up additional space in the paper so it still fits) and add additional comments therein. Unfortunately, it is now past the time that we can make the modification.
> > >
> > > Regarding Table 3, this experiment only uses Auto-Transfer (route) and shows a comparison between the performances of Finetuning and Auto-Transfer (route) using identical architectures. This experiment was suggested by Reviewer  3TiT and only uses route in order to demonstrate the simplest baseline transfer (using source and target with identical architectures). We apologize for the confusion and note that our comment titled "General Comments" lists all of our modifications and correctly identifies the new tables and figures.
> > >
> > > Please let us know what other concerns you may have - we are happy to discuss.

---

### Official Review · Reviewer_u325 · 2021-11-01

**Correctness:** 3
**Technical Novelty And Significance:** 3
**Empirical Novelty And Significance:** 3
**Recommendation:** 6
**Confidence:** 5

**Main Review:**


1. The method may be not novelty enough. It tries to introduce reinforcement learning into transfer learning to route transferable representations. The reinforcement learning part utilizes a traditional bandit-based method while routing choices are common choices that have been used in existing work widely.
2. It is better to add more latest state-of-the-art comparison methods. This paper only compares with methods used in L2T-ww. However, L2T-ww is a paper proposed in 2019, which cannot represent the best results now. Some more work, such as SNOW[1], has been proposed for knowledge transfer.
3. Since the paper is similar to L2T-ww, a method proposed in 2019, it is better to compare all results showed in L2T-ww. And more new benchmarks are needed.

Overall, I recommend rejecting the current version of the paper.

[1] Yoo C, Kang B, Cho M. SNOW: Subscribing to Knowledge via Channel Pooling for Transfer & Lifelong Learning of Convolutional Neural Networks[C]//International Conference on Learning Representations. 2020.

**Summary Of The Paper:**

To transfer knowledge between heterogeneous source and target networks and tasks, this paper proposes a novel adversarial multi-armed bandit approach (AMAB) which automatically learns to route source representations to appropriate target representations. It combines feature representations received from the source network with the target network-generated feature representations via various aggregation operations. The work is interesting and makes sense to some extent.


**Summary Of The Review:**

See main review.

---

> ### Author Response · Authors · 2021-11-18
> **Response to Reviewer u325**
>
> We thank the reviewer u325 for helping improve our work. We address the major concerns below,
>
> **Novelty**
>
> #### Q1. The method may be not novelty enough. It tries to introduce reinforcement learning into transfer learning to route transferable representations. The reinforcement learning part utilizes a traditional bandit-based method while routing choices are common choices that have been used in existing work widely.
>
> A1. To the best of our knowledge, we did not find any earlier works in transfer learning literature that utilizes or mentions routing in their proposed approach.
>
> **Comparison to latest SoTA methods**
>
> #### Q2. It is better to add more latest state-of-the-art comparison methods. This paper only compares with methods used in L2T-ww. However, L2T-ww is a paper proposed in 2019, which cannot represent the best results now.
>
> A2. As suggested by the reviewer, we found the most recent baseline from AAAI 2021 that improves upon L2T-ww to be SAaD[1]. As seen in the table below, we find that our method outperforms SAaD in all benchmark datasets. We have also updated Table 1 with the SAaD baseline.
>
> |                       | CUB200            | Stanford Dogs     | MIT67             | Stanford40        |
> |-----------------------|-------------------|-------------------|-------------------|-------------------|
> | SAaD [1]               | 68.29&pm;DNR      | 76.06&pm;DNR      | 66.47&pm;DNR      | 63.08&pm;DNR      |
> | Auto-Transfer (route) | **74.76**&pm;0.39 | **86.16**&pm;0.24 | **75.86**&pm;1.01 | **80.10**&pm;0.58 |
> *DNR: did not report
>
> #### Q3. Since the paper is similar to L2T-ww, a method proposed in 2019, it is better to compare all results showed in L2T-ww. And more new benchmarks are needed.
>
> A3. As seen in Table 1, we already compare our method against ImageNet based transfer from L2T- ww. We add the TinyImageNet based transfer results as well and find that our method performs better in comparison to L2T-ww (Table 6 in Appendix A.5).
>
> ---
> References:
>
> [1] Show, Attend and Distill: Knowledge Distillation via Attention-based Feature Matching, AAAI 2021

---

> > ### Comment · Reviewer_u325 · 2021-11-22
> > **Thank you for your response**
> >
> > I appreciate the results and responses from the authors. Much of my concerns are resolved. I would like to increase my score to 6.

---

### Official Review · Reviewer_7soo · 2021-11-03

**Correctness:** 2
**Technical Novelty And Significance:** 3
**Empirical Novelty And Significance:** 2
**Recommendation:** 6
**Confidence:** 3

**Main Review:**

Strengths:
* I like the principled approach to letting the model figure out what information to transfer. The additional inductive bias of constraining the information flow could be useful to directly inspect what information is being transferred between the networks.

Weaknesses:
* Evaluation setup: is there a standard benchmark for this task? And more recent baselines? If your interest in this setup is driven by addressing scenarios with limited amounts of labels, more careful evaluation of this scenario would be meaningful to add.
* Ablation studies: it would be interesting to investigate deeper the effect of choosing adversarial bandits in this context. The Appendix reports an ablation study with a fixed routing choice, but other baselines could be interesting too. What's the effect of the frequency of updates of the bandit? How do different parts of the proposed method impact the overall effectiveness? How does the choice of number of representations influence the performance? Are these learned intermediate representations meaningful?
* The authors mention the ability to decide whether to overwrite the target's network own information as an advantage of their method. Are there ways to test this hypothesis? It seems like giving the option to let the network choose the aggregation operation resulted in always the same choice,

Minor:
* Some experimental details are missing, for example how many random initializations where tried to compute the variance over the reported accuracy?
* Time improvements: how do the results depend on the complexity of the routing problem?


**Summary Of The Paper:**

This work addresses the challenge of automatic knowledge transfer between different networks. The authors propose to use an adversarial  multi-armed bandit to decide where, what and how to combine outputs from different layers of the two networks, improving on the recent line of work that achieves transfer by enforcing closeness between the representations obtained by the two networks.

For each layer, the proposed method introduces some additional parameters in the form of intermediate representations and parametrized ways to combine such intermediate representations with the source network's output.  The multi-armed bandit is trained to choose which intermediate representation and which aggregating function to use. The reward is determined based on a hold out set. The authors compare the proposed approach with some previous work, showing improvements, and perform a qualitative analysis of the results.

**Summary Of The Review:**

While the authors address an important problem with an interesting technique, a more principled evaluation would be necessary for this work to be complete. The task should be better contextualized within the existing literature, and the proposed method evaluated more throughly.

---

> ### Author Response · Authors · 2021-11-18
> **Response to Reviewer 7soo**
>
> We thank reviewer 7soo for helping improve our work. We address the major concerns below.
>
> **Evaluation setup**
>
> #### Q1. is there a standard benchmark for this task? And more recent baselines? If your interest in this setup is driven by addressing scenarios with limited amounts of labels, more careful evaluation of this scenario would be meaningful to add.
>
> A1. The four datasets in our experimental evaluation are standard benchmarks used in recent transfer learning literature [1,2,3]. The most recent baseline (from AAAI 2021) that we can compare to is SAaD [3]. We have also updated Table 1 with the SAaD baseline. Our method significantly outperforms SAaD[3] as seen in the table below.
>
> |                       | CUB200            | Stanford Dogs     | MIT67             | Stanford40        |
> |-----------------------|-------------------|-------------------|-------------------|-------------------|
> | SAaD [3]               | 68.29&pm;DNR      | 76.06&pm;DNR      | 66.47&pm;DNR      | 63.08&pm;DNR      |
> | Auto-Transfer (route) | **74.76**&pm;0.39 | **86.16**&pm;0.24 | **75.86**&pm;1.01 | **80.10**&pm;0.58 |
> *DNR: did not report
>
> We address limited data scenarios in our work by training Auto-Transfer with 10% to 100% of training data (in 10% increments). Results from this study on Stanford40 dataset can be found in Figure 2 (top). We repeated the same experiment with three other datasets and found similar results. These new results are added in Figure 9 (Section A.5) and [here](https://imgur.com/a/fAjLQhc). Additionally, we report experiments on smaller datasets such as CIFAR100 and STL10 using TinyImageNet-based transfer (Table 6 in Section A.5).
>
> **Ablation studies**
>
> #### Q2. it would be interesting to investigate deeper the effect of choosing adversarial bandits in this context ...
>
> A2. We believe we tried the most important ablation amongst the ones the reviewer suggested. In particular, we already report an ablation that demonstrates the importance of adversarial bandits (Table 7 and please see our response A4 to reviewer 3TiT). Next, we demonstrate the effect of different aggregation operators in Table 8. In terms of frequency update, we tried the standard exp setting of 1 update per epoch. Further improvement may be achieved by experimenting with several setup. However even with this standard experimental setup, we have shown significantly better performance than the state-of-the-art.
>
> We show an example of intermediate representation learned by Auto-Transfer (route) method in Figure 10 and [here](https://imgur.com/a/7qbFKdA). Although some patterns can be seen in these intermediate representations, Grad-CAM gives more insightful details about which parts of the input image are important. We demonstrate through several Grad-CAM based visual explanations (Figures 3, 11, 12, 13) that our method successfully identifies relevant pixels for input images.

---

> > ### Author Response · Authors · 2021-11-18
> > **Response to Reviewer 7soo (Continued)**
> >
> > **Aggregation of representations**
> >
> > #### Q3. The authors mention the ability to decide whether to overwrite the target’s network own information as an advantage of their method. Are there ways to test this hypothesis? It seems like giving the option to let the network choose the aggregation operation resulted in always the same choice
> >
> > A3. We apologize for the confusion. As described in the paper, when the network is allowed to choose an aggregation operator (Auto-Transfer (full)), it ends up choosing Weighted Addition (wtadd) most of the times. Taking this into account, we then repeated the experiments by fixing the aggregation to Weighted Addition (Auto-Transfer (route)). Even in this case, the network has the liberty to choose the amount of source and target representations to combine since a scalar weight is learned  and multiplied to them.
> >
> > $\tilde f_{T}^i = w_{S,i,j} * \tilde f_{S}^j + w_{T,i,j} * f_{T}^i$
> >
> > In terms of validating the usefulness of this, let us consider a simple case of using Simple addition (sAdd) vs Weighted addition (wtAdd) for aggregation operation. In sAdd, both source and target representations are added together. With wtAdd, each representation is multiplied by a learned scalar parameter that controls for how much source and target representation to use. Below we show the results for the Stanford40 dataset,
> >
> > |               | SAdd          | WtAdd             |
> > |---------------|---------------|-------------------|
> > | Auto-Transfer | 77.78&pm;1.18 | **80.10**&pm;0.58 |
> >
> > We see that WtAdd performs better in comparison to sAdd showing that it’s useful to let the target network choose proportion of source and target representations. Also, consider this example of scalar weights associated with wtAdd operation for each target layer receiving representation from source layer 1,
> >
> > |          | source weight | target weight |
> > |----------|--------|--------|
> > | Target 1 | 0.42   | 0.57   |
> > | Target 2 | 0.32   | 0.66   |
> > | Target 3 | 0.22   | 0.73   |
> > | Target 4 | 0.08   | 0.99   |
> >
> > We see that the amount of target representation increases as we get deeper in the network. This is expected since the representations become task specific as we go deeper in the network.
> >
> > Finally, we would like to point that in the case of TinyImageNet to CIFAR100 transfer (Table 6), Auto-Transfer (full) performed better than Auto-Transfer (route) indicating that wtAdd is not necessarily the operation chosen for best performance in all scenarios.
> >
> >
> > **Minor comments**
> >
> > #### Q4. how many random initializations where tried to compute the variance over the reported accuracy?
> >
> > A4. We thank the reviewer for pointing this out. All experiments were repeated 3 times. We have updated the manuscript to reflect this.
> >
> > #### Q5. Time improvements: how do the results depend on the complexity of the routing problem?
> >
> > A5.We did a thorough analysis of training/inference time vs performance of our method in comparison to Finetuning. Additional details can be found in our responses to Reviewer 3TiT (A2, A3).
> >
> > ---
> > References:
> >
> > [1] DELTA: Deep Learning Transfer using Feature Map with Attention for Convolutional Networks, ICLR 2019
> >
> > [2] Pay Attention to Features, Transfer Learn Faster CNNs, ICLR 2020
> >
> > [3] Show, Attend and Distill: Knowledge Distillation via Attention-based Feature Matching, AAAI 2021

---

> > > ### Author Response · Authors · 2021-11-25
> > > **Minor correction on our previous response**
> > >
> > > In our answer (A3), we accidentally wrote Table 3 instead of Table 6. We have fixed this in the above response.

---

### Official Review · Reviewer_3TiT · 2021-11-05

**Correctness:** 4
**Technical Novelty And Significance:** 3
**Empirical Novelty And Significance:** 3
**Recommendation:** 6
**Confidence:** 4

**Main Review:**

Transfer learning is becoming an important tool for applied machine learning. On one hand, it can help to save lots of energy and compute by training once and benefitting many times from it. On the other, it can unlock the use of small datasets that, in isolation, are hard to crack. Accordingly, any development in this front can have significant consequences in the practical use and adoption of modern deep learning (or ML, more broadly). So far, the golden standard is to finetune the model on the new dataset (maybe after distilling to the target architecture if needed, and possibly by applying some additional regularization to avoid overfitting or diverging too much from the original model).

This paper proposes an alternative. There are many ways to use the source predictions and representations in a potentially useful downstream manner. Rather than choosing one, and just committing to it in all cases (e.g., full finetuning), they propose an adaptive mechanism that operates in two dimensions. First, it essentially considers a set of architectures (one for each potential source-target plugging). Second, for a given wiring, it trains part of it (all except the original source network which remains frozen) on the downstream data. The former is a discrete optimization problem, the latter is a continuous one. A bandit algorithm (Algorithm 1) and gradient descent (Algorithm 2) are used respectively, in an alternating fashion.

In my opinion, the idea is quite natural. There are some subtleties, though. Algorithm 2 changes the "environment" by training, so it's a non-stationary problem. Also, rather than considering the joint (or say, Cartesian product) action space, the algorithm solves one bandit problem per target layer as an independent instance. The reward designed by the algorithm (given in Algorithm 3, line 2) assumes no "decision" is applied in the other bandits problems, and tries to find the best action for the layer while assuming there won't be any wiring in the remaining target layers (if I understand correctly).

I find the experimental section to be a bit limited.

First, the simplest baseline I can think of (source and target identical architectures, and finetune source into target) is not there: ResNet X to ResNet X. There's a comparison ("Scratch (finetune)") where a smaller source network (RN18) is used to help the target (RN18), but still compared with Auto-Transfer on a RN36 source network, so not exactly apples to apples. It would be extremely informative to see this comparison as I think it is the cleanest one. Moreover, the direct finetuning should be much faster; can we also see a time-matched comparison (i.e. train for the same amount of time the RN-X to RN-X finetune transfer and the Auto-Transfer approach)? I'll raise my score if this is added (and given the numbers in Table 1, I'd be confident this should still work well for Auto-Transfer).

The numbers in Table 1 are really impressive: Auto-Transfer seems to offer massive gains. There's only 4 datasets though, considering something like VTAB [1] could improve the work (it contains a large number of diverse downstream vision tasks). Also, it'd be interesting to see a more detailed ablation to understand which aspects of the algorithm really explain the gains. Is the bandit algorithm fundamentally important? Can we use a simple UCB? Or even an epsilon greedy approach based on the average reward? Or is it mostly about the action set --some of this is shown in the full/fixed/route comparison? For example, rather than applying the (somewhat complex) Auto-Transfer algorithm every time we want to transfer, a more desirable outcome would be to discover specific combinations (i.e. actions in the action set) --that is, transfer design principles-- that work well across many different setups. Maybe we can look at the finally selected ones for the datasets in Table 1, find a simple common pattern (i.e. one choice of the action set), and train with this fixed choice, and see how it does? In those cases, maybe rather than the algorithm, the contribution could also include some high level wiring tips (this may seem at first less appealing as it's tailored towards one specific architecture: resnet, transformer, etc) but it could have a much bigger impact.

A potential explanation to beating simple finetuning may have to do with the intrinsic memory that this algorithm provides. In some sense, by paying in terms of inference cost and storage, we can't forget the original representations. The final models have very different size/dimension/power (if you keep the source model or not around).

This leads to another big question: inference. Regardless of the cost and complexity of finetuning, the proposed approach requires that during inference we'll need to apply both networks (source and target). Moreover, in general sequential processing is needed: you have to run source first, as its last layer representation could be wired to the first target layer one. Can you provide results where the inference time is matched (ignoring complexity)? In other words, compare a larger model directly finetuned to, with two smaller source and target networks trained via Auto-Transfer, so that total inference flops or runtime in both cases are identical. At the end of the day, these considerations probably matter a lot in real practical systems. Looking at Figure 2, I could be optimistic about these cost-matched comparisons, if one model is able to unlock much higher performance values, then it seems no matter how much time you devote to the suboptimal one, it may not be able to catch up.


[1] = A Large-scale Study of Representation Learning with the Visual Task Adaptation Benchmark

**Summary Of The Paper:**

The paper proposes an algorithm to transfer a "source" pre-trained deep model into a new one for a given target task. The idea is to wire in a certain way the intermediate representations of the source model into those of the "target" one. The algorithm considers a (small) discrete space of potential such ways to connect both models (location, source processing, and type of merging), and applies a bandit algorithm to sequentially spend training budget under one of the configurations. Experiments are provided that show big gains with respect to standard approaches (like finetuning the source model).

**Summary Of The Review:**

The paper proposes a sequential decision making idea to do transfer learning using and keeping two networks: pre-trained source and from-scratch target. The results that are presented suggest Auto-Transfer offers large gains, but the scope of the experimental section is a bit limited.

--------------

After reading the detailed reply by the authors, I decided to raise my score: 5 --> 6.

---

> ### Author Response · Authors · 2021-11-18
> **Response to Reviewer 3TiT**
>
> We thank the reviewer 3TiT for helping to improve our work. We address the concerns below.
>
> **Training performance with identical source and target architectures**
>
> #### Q1. First, the simplest baseline I can think of (source and target identical architectures, and finetune source into target) is not there: ResNet X to ResNet X.
>
> A1. As suggested by the reviewer, we report a comparison between Finetuning and AutoTransfer (route) using matched architecture (i.e., ResNet18 - ResNet18). Please find the performance table below,
>
> |             |       CUB200      |   Stanford Dogs   | MIT67             | Stanford40        |
> |-----------|-----------------|-----------------|-------------------|-------------------|
> | FT(18 - 18) |   42.96&pm;1.45   |   53.02&pm;3.57   | 47.93&pm;3.66     | 34.40&pm;5.94     |
> | AT(18 - 18) | **66.97**&pm;1.38 | **79.46**&pm;1.05 | **69.54**&pm;2.49 | **75.07**&pm;2.55 |
>
>
>  From the above table, we find similar performance gains for the Auto-Transfer with the identical architectures.
>
>  #### Q2. Can we also see a time-matched comparison (i.e. train for the same amount of time the RN-X to RN-X finetune transfer and the Auto-Transfer approach)?
>
> A2. We have also added time-matched training curves comparing Finetuning and Auto- Transfer (route) for matched architectures (i.e., ResNet18 - ResNet18). Please find them in Figure 7 (Section A.4) and [here](https://imgur.com/a/Jme0vTz). We find that our method significantly outperforms Finetuning when trained for same amount of time.
>
> **Inference on time matched architectures**
>
> #### Q3. Can you provide results where the inference time is matched (ignoring complexity)? In other words, compare a larger model directly finetuned to, with two smaller source and target networks trained via Auto-Transfer, so that total inference flops or runtime in both cases are identical.
>
> A3. As suggested by the reviewer, we report the inference times for batches of 128 images and compare Finetuning (ResNet34 - ResNet34) vs AutoTransfer (ResNet18 - ResNet18) . These models: larger models with Finetuning versus smaller models with AutoTransfer, take up approximately the same amount of disk space and inference times. We also report AutoTransfer (ResNet34 - ResNet18) for comparision. We find that the Auto-Transfer (ResNet18 - ResNet18) method significantly outperforms Finetuning (ResNet34 - ResNet34) while taking similar amount of disk space and inference time. AutoTransfer (ResNet34 - ResNet18) improves performance further while taking about 4 seconds more (i.e., approximately 31 milliseconds per image). We show the average time taken vs average test accuracy in the table below while the scatter plots with individual datapoints can be found in Figure 8 (Section A.4) and [here](https://imgur.com/a/xOUtmMo).
>
> |             |   CUB200   |         | Stanford Dogs |         | MIT67      |         | Stanford40 |         |
> |:-----------:|:----------:|---------|:-------------:|---------|------------|---------|------------|---------|
> |             | time (sec) | Acc (%) | time (sec)    | Acc (%) | time (sec) | Acc (%) | time (sec) | Acc (%) |
> | FT(34 - 34) |    12.88   | 37.13   |     12.66     | 52.26   | 12.22      | 44.37   | 14.0       | 31.12   |
> | AT(18 - 18) |    14.46   | 64.37   |     13.83     | 77.07   | 14.26      | 67.89   | 15.28      | 69.02   |
> | AT(34 - 18) | 18.55      | 71.84   | 18.27         | 85.09   | 18.62      | 69.76   | 19.20      | 79.74   |

---

> > ### Author Response · Authors · 2021-11-18
> > **Response to Reviewer 3TiT (Continued)**
> >
> > **Ablation to understand the gains**
> >
> > #### Q4. Is the bandit algorithm fundamentally important? ... Or is it mostly about the action set? ... Maybe we can look at the finally selected ones for the datasets in Table 1, find a simple common pattern (i.e. one choice of the action set), and train with this fixed choice, and see how it does?
> >
> > A4. As described in section 3.2 *Environment update*: In our non-stationary problem setting, the knowledge transfer from the source model changes the best action (and the reward function) at every round as the target network adapts to this additional knowledge. This is the key reason to use adversarial bandits for making choices as it is agnostic to an action dependent adversary.
> >
> > In terms of finding common patterns among action choice, we found that the final layer pairs selected by Auto-Transfer (route) for Table 1 results are not uniform across multiple repetitions and different datasets. Please find the selected source layer for each target layer listed below (experiments repeated 3 times and each cell is written as source layer selected at: run1, run2, run3),
> >
> > | Target Layer  | Layer 1 | Layer 2 | Layer 3 | Layer 4 |
> > |---------------|---------|---------|---------|---------|
> > | CUB200        | 2, 2, 2 | 3, 2, 2 | 2, 1, 1 | 2, 4, 4 |
> > | Stanford Dogs | 1, 1, 4 | 3, 3, 5 | 2, 3, 2 | 4, 5, 4 |
> > | MIT67         | 2, 4, 2 | 3, 1, 5 | 2, 3, 1 | 3, 3, 4 |
> > | Stanford40    | 1, 2, 4 | 4, 3, 3 | 2, 3, 2 | 3, 4, 3 |
> >
> > In order to test our model further, we trained Auto-Transfer (fixed, retrain), with the selected pairs above and found that the performance is worse in comparison to training with bandits (i.e., Auto- Transfer (route)). We reported this ablation study in our original submission (Section A.6).
> >
> >
> > **Experiments on additional datasets**
> >
> > #### Q5. The numbers in Table 1 are really impressive: Auto-Transfer seems to offer massive gains. There’s only 4 datasets though, considering something like VTAB [1] could improve the work (it contains a large number of diverse downstream vision tasks).
> >
> > A5. Thank you for appreciating the efficacy of our proposed method. Our selection of benchmark datasets replicates what has been done in recent state-of-the-art transfer methods that we compare against (Table 1). Adapting the code of previous works for new benchmarks is difficult due to long run-times of meta learning methods such as L2T-ww. Also, we note that the focus of VTAB is on representation learning rather than transfer learning, and it is not obvious what source datasets to use for certain tasks (namely among the Specialized and Structured tasks of VTAB). Nevertheless, we have added additional experiments for TinyImageNet based transfer on CIFAR100 and STL10 datasets (Table 6).

---

> > > ### Comment · Reviewer_3TiT · 2021-11-30
> > > **Thank you for your reply.**
> > >
> > > I'd like to thanks the authors for the detailed reply and new results, tables, and plots.
> > >
> > > I find the answers and results quite convincing, and accordingly I'd like to raise my score.

---

### Author Response · Authors · 2021-11-18
**General Comments**

We thank all the reviewers for the detailed feedback and suggestions. We have tried to address many of your comments leading to changes in the submission. including several experiments, that we list here so that all reviewers are aware of the updates:

 - *Additional Baselines:*  We have done additional analysis on the training and inference performances for Auto-Transfer  and added in Appendix Section A.4.
    - In Table 1, we have added the most recent baseline SAaD from AAAI 2021 [1].
    - Figure 7 and Table 3 show comparison between the performances of Finetuning and Auto-Transfer (route) using identical architectures. We see that when trained for the same amount of time Auto-Transfer performs significantly better than Finetuning.
 - *Additional Benchmark Datasets:* Table 6 shows additional experimental results on transfer from TinyImageNet to CIFAR100 and STL10. We see improvements over the other baselines.
 - *Additional Ablation Study:*
      - Inference Time Analysis: Figure 8 and Table 4 compare the time-matched inference performances of Auto-Transfer and Finetuning. We show that Auto-Transfer outperforms Finetuning while taking up approximately the same amount of disk space and inference times.
     - Limited Label setting: We have added additional limited data experiments for CUB200,  Stanford Dogs, Mit67 in Figure 9.
     - We have done additional ablation on fixing different aggregation operators (Iden, SAdd, WtAdd, LinComb, FactRed) and show that WtAdd leads to best performance and shown in Table 8.
     - We have added the example of learned intermediate representations for a bird image from CUB200 dataset for Auto-Transfer in Figure 10 Section A.7.

We have revised our paper to reflect the aforementioned changes.

 ---
Reference:

[1] Show, Attend and Distill: Knowledge Distillation via Attention-based Feature Matching, AAAI 2021

---

### Author Response · Authors · 2021-11-22
**Checking in**

We again thank the reviewers for their detailed comments. Today is the last day that we are allowed to update the submission. We request the reviewers to read the general comments and individual responses where we have reported our additional experiments and addressed specific questions. If a reviewer has any suggestion that requires further changing the submission, please let us know.

---

### Author Response · Authors · 2021-12-08
**Thank you for all the suggestions**

Dear Reviewers,

We are glad that you have all increased your scores and find our answers and additional results to be helpful. We are grateful for the suggestions that have led to an improved paper.

Thank you,

Authors

---

### Decision · Program_Chairs · 2022-01-20

**Decision:**

Accept (Poster)

**Comment:**

This paper tackles the problem of how to utilize a network from the source domain to benefit target domain training in terms of sample/training efficiency. In contrast to prior methods (e.g. that perform fine-tuning or distillation), this paper poses it as a bandit problem that decides how to wire intermediate representations of the source model into the target model as well as what aggregation function to use. An alternating/mixed discrete-continuous optimization is proposed to perform this decision-making, and results are shown across a mix of source-target pairs and network architectures.

  The reviewers overall found the method interesting and paper topic both interesting and extremely practically useful, presenting an opportunity to save significant energy, compute, and labeling requirements when training on target domains. The results also show very significant improvements, on the conditions tried. However, a number of concerns were raised including comparison to simple same-architecture fine-tuning (3TiT), comparison benchmarks e.g. VTAB and newer methods in the area (u325, 7soo, 3TiT), need for the adversarial bandit formulation (3TiT, 7soo, 8d9w), and the added storage/inference costs required (3TiT).

  Based on these reviews, the authors provided a thorough rebuttal, additional baselines and experiments demonstrating the efficacy of the method (especially the full version) over both reasonable simple baselines (same architecture fine-tuning) as well as simpler versions (fixed aggregation), and time/inference-time matched performances. Importantly, the advantage of the full method comes out a lot more in the new experiments. Overall, through the rebuttal process the paper has been made much stronger.

  Given that the paper provides a nice principled approach to the problem and now has strong compelling results, I recommend acceptance.